# Gastruloids are competent to specify both cardiac and skeletal muscle lineages

Laurent Argiro [1,3], Céline Chevalier [1,3], Caroline Choquet[1,3], Nitya Nandkishore[1,2], Adeline Ghata[1], Anaïs Baudot [1], Stéphane Zaffran [1,4] ✉ & Fabienne Lescroart [1,4] ✉

Cardiopharyngeal mesoderm contributes to the formation of the heart and head muscles. However, the mechanisms governing cardiopharyngeal mesoderm specification remain unclear. Here, we reproduce cardiopharyngeal mesoderm specification towards cardiac and skeletal muscle lineages with gastruloids from mouse embryonic stem cells. By conducting a comprehensive temporal analysis of cardiopharyngeal mesoderm development and differentiation in gastruloids compared to mouse embryos, we present the evidence for skeletal myogenesis in gastruloids. We identify different subpopulations of cardiomyocytes and skeletal muscles, the latter of which most likely correspond to different states of myogenesis with "head-like" and "trunk-like" skeletal myoblasts. In this work, we unveil the potential of gastruloids to undergo specification into both cardiac and skeletal muscle lineages, allowing the investigation of the mechanisms of cardiopharyngeal mesoderm differentiation in development and how this could be affected in congenital diseases.

Deployment of progenitors and their proper allocation to correct cell lineages are fundamental for the formation of organs. Defects in the specification and differentiation of progenitors into particular cell lineages lead to congenital anomalies. For example, a recent report showed improper cardiopharyngeal mesoderm progenitor development in *Tbx1* conditional mutant embryos, a widely used 22q11.2 deletion syndrome (DS) mouse model, where head muscle and heart morphogenesis were impaired[1].

The heart forms from two cell populations, namely the first and second heart fields. The first heart field forms essentially the left ventricle with small contributions to the right ventricle and atria[2,3]. Second heart field progenitors, located in cardiopharyngeal mesoderm (CPM), contribute to cardiac muscles (myocardium) of the outflow tract, right ventricle and atria[2–7]. Cells within the CPM populates the pharyngeal arches and also gives rise to head and a subset of neck branchiomeric skeletal muscles, in addition to cardiac muscles[8–11]. Clonal analyses in the mouse model have revealed the

existence of bipotent progenitors in CPM that form both head branchiomeric and heart muscles in vertebrates with different contributions along the antero-posterior axis of the pharyngeal region of the embryo[8,12,13]. Bipotent progenitors were thus found for distinct subsets of myocardium and specific head and neck skeletal muscles[12,13]. Lineage tracing of early nascent mesoderm expressing the bHLH transcription factor Mesp1 showed that bipotent head and heart muscle progenitors are in fact present at the onset of gastrulation[14]. CPM is evolutionarily conserved since similar multipotent cardiopharyngeal progenitors have been shown to give rise to heart and pharyngeal muscles in tunicates[9,15].

For its skeletal muscles derivatives, CPM contributes to specific branchiomeric head/neck muscles[9,10]. Skeletal muscles derived from CPM have distinct genetic regulatory programs compared to trunk and limb skeletal muscles[16]. Trunk and limb skeletal muscles derive from somitic paraxial mesoderm. Somitic myogenesis is controlled by Pax3 (a homeobox paired-domain transcription factor), which regulates the

---

[1]Aix-Marseille Univ, INSERM, Marseille Medical Genetics (MMG), Marseille, France. [2]Present address: Department of Biotechnology, SRM Institute of Science and Technology (SRMIST), Kattankulathur, Tamil Nadu, India. [3]These authors contributed equally: Laurent Argiro, Céline Chevalier, Caroline Choquet. [4]These authors jointly supervised this work: Stéphane Zaffran, Fabienne Lescroart. ✉e-mail: stephane.zaffran@univ-amu.fr; fabienne.lescroart@univ-amu.fr

myogenic regulatory transcription factors MyoD and Myf5. Strikingly, *Pax3/Myf5* mutant embryos failed to develop trunk and limb muscles while head muscles were broadly not affected, thus decoupling trunk (somitic) from head (cardiopharyngeal) myogenesis[17]. In CPM, the myogenic factors MyoD/Myf5 are regulated by a different core of transcription factors that together mark the CPM but are individually not strictly restricted to this population. They include Tbx1, Lhx2, MyoR (also called Msc), Tcf21 (also called Capsulin), and Pitx2[18–22]. Myogenin (MyoG) then marks the start of myoblast differentiation. Pax7 is later involved in both CPM and somitic skeletal muscles[23]. The next steps of muscle differentiation are common between cardiopharyngeal and somitic mesoderm derived progenitors[16].

Major insights into the specification of multipotent CPM progenitors and the mechanisms of cardiopharyngeal lineage segregation have come from studies in *Ciona intestinalis*. In this tunicate model, Tbx1/10 activates COE/Ebf for the specification of the muscular lineage[24,25]. It is however not completely clear how comparable are tunicates and vertebrates. The heart and pharyngeal muscles derive from only 2 Mesp+ progenitors in tunicates, while about 250 Mesp1+ progenitors contribute to heart development in mouse[9,26]. Despite the number of important studies using lineage tracing and clonal analysis, it is challenging to analyze CPM specification in vertebrates due to its difficult accessibility and the absence of restricted and specific markers. There is a need of a simple model that could faithfully recapitulate vertebrate, and more specifically mammalian, CPM development and could allow live-imaging and large throughput analyses.

Pluripotent stem cells have emerged as an interesting tool in general to model how transcription factors and signaling molecules interact to control cell fate decisions and lineage specification[27,28]. Using mouse embryonic stem cells (mESCs), Chan et al. have shown that Mesp1-derived progenitors have a dual heart and skeletal muscle differentiation potential, when cultured in a pro-cardiogenic or a pro-skeletal myogenic culture medium[29]. Mouse and human ESCs have also been used to investigate the differences in molecular cues involved in myogenic specification between cardiopharyngeal versus somitic mesoderm[30]. Both studies however, required changes in cell culture medium and growth factors to promote either cardiac or skeletal muscle differentiation[29,30]. Therefore, there is a lack of an in vitro model that allows parallel differentiation of cardiac and skeletal muscle cells and faithfully recapitulates in vivo CPM early development.

The establishment of gastruloids provides an alternative model for CPM development. The model of gastruloids was first described in 2014 with axial elongation and symmetry breaking from the aggregation of a restricted number of mESCs cultured with a pulse of Wnt activation (with Chiron treatment)[31,32]. This model faithfully recapitulates early mouse development mimicking in vivo gene expression[33]. Recently, the model has been pushed further towards early organogenesis, with development of heart-like structures with first and second heart field derivatives[34,35] or somites[36,37]. Despite the formation of somites, skeletal myogenesis, either cardiopharyngeal or somitic, has not yet been demonstrated in gastruloids.

Here, we have adapted the self-organizing mESCs-based gastruloid protocol for longer times of culture to investigate whether gastruloids can model CPM specification and differentiation. Multiplex fluorescent in situ hybridization shows that CPM specification occurs in gastruloids in a similar spatio-temporal organization as observed in mouse embryos. Using single-cell RNAseq analysis along a time-course from day 4 to late day 11 of culture, we demonstrate the presence of three subpopulations of cardiomyocytes and two subpopulations of myoblasts. We find that gastruloids can undergo myogenesis and that gastruloid-derived myoblasts can arise from both CPM and somitic paraxial mesoderm. Therefore, gastruloids can be used to reconstruct CPM development.

## Results

### CPM markers are expressed in gastruloids

To generate gastruloids that form CPM and characterize their differentiation to cardiac and skeletal myogenic fates, we extended a previous protocol described by Rossi et al.[35] to culture gastruloids for an prolonged time, until day 11 (Fig. 1a) (see Methods). In brief, mESCs were aggregated at day 0, following centrifugation, and treated with a Wnt agonist (Chiron) for 24 h from day 2. Cardiogenic factors (bFGF, VEGF and ascorbic acid) were added to the culture media at 96 h (day 4) for 3 days. After day 7, gastruloids were cultured in N2B27 culture media. Shaking (80–100 rpm) was continuous from day 4 to the end of the culture. As previously reported[32,35,38], we observed efficient and robust elongation starting on day 4, with beating areas appearing by day 7. On average, about 86.79% (± 7.4 SEM) of gastruloids in each experiment showed beating areas, primarily in the anterior part of the gastruloid (Fig. 1b, c and Supplementary Video 1). On day 7, gastruloids express cardiac troponin T (cTnT), VEGFR2 and E-cadherin, as previously reported[35], indicating their ability to form cardiomyocytes, endothelial cells and endoderm (Supplementary Fig. 1a, b). Interestingly, gastruloids on day 11 continue to express cTnT, VEGFR2 and E-cadherin (Supplementary Fig. 1c, d).

To investigate whether the anterior-posterior axis is maintained at day 11, we used multiplex fluorescent in situ hybridization or RNAscope to analyze the expression of *Hoxc4*, a posterior marker[39]. We found *Hoxc4* expression at one pole of the gastruloids at both day 7 and day 11 (Supplementary Fig. 1e, f). However, we observed an anti-correlation between the expression of *Hoxc4* and *troponin (Tnnt2)*, a marker of cardiomyocytes, where the expression of *Hoxc4* decreased as *Tnnt2* expression expanded. This suggests that as gastruloids develop more cardiomyocytes, they lose their posterior region, with highly beating gastruloids becoming predominantly anterior. With the extension of the protocol, gastruloid morphology is affected from day 7.

To investigate whether key markers of CPM and its derivatives (Fig. 1d) were expressed during the time-course culture, we used quantitative RT-PCR. We observed the transient expression of the bHLH transcription factor *Mesp1* and the expression of the CPM transcription factors *Islet1 (Isl1)* and *Tbx1*, as previously described[35] (Fig. 1e). We also observed an increasing expression of transcripts encoding the cardiac specific myosin (*Myl7 and Myh7*) and *Tnnt2* at day 5 (Fig. 1f). Interestingly, the CPM marker, *Tcf21*[20,25], began to be expressed at day 3, while the myogenic transcription factors *Myf5* and *MyoD* were expressed at day 7 (Fig. 1g). Importantly, similar kinetics of expression were also observed for *Mesp1*, *Isl1* and *Tcf21* with another wild-type mESC line (see Methods for details on the lines and Supplementary Fig. 2). These data demonstrate that CPM and downstream myogenic transcriptional programs are robustly activated in gastruloids under these extended culture conditions.

### Similar spatio-temporal gene expression with mouse embryo

To explore whether gastruloids faithfully mimic mouse CPM development, we compared gastruloids with mouse embryos at equivalent developmental stages. Our goal was to investigate the expression of markers of the CPM, cardiomyocytes, and the robustness of the CPM specification. We used multiplex fluorescent in situ hybridization (RNAscope or HCR) to explore gene expression. We first compared early day 4, spheric, gastruloids with E7.5 (Early bud stage) embryos. It showed a small overlapping expression of *Mesp1* with *Isl1* and in a smaller proportion with *Tbx1* (Fig. 2a–a' and c–c'). After symmetry breaking, comparison of day 5 gastruloids with E7.75 cardiac crescent embryos showed that *Mesp1* expression is in non-overlapping domains with *Isl1* and *Tbx1* (Fig. 2b–d and Supplementary Fig. 3). In the embryo, *Mesp1* was observed in the posterior mesoderm (likely corresponding to the somitic mesoderm), while *Tbx1* and *Isl1* expression overlapped in the anterior region of the embryo (Fig. 2b). Similarly, *Tbx1* and *Isl1* expression overlapped in gastruloids (Fig. 2d–d').

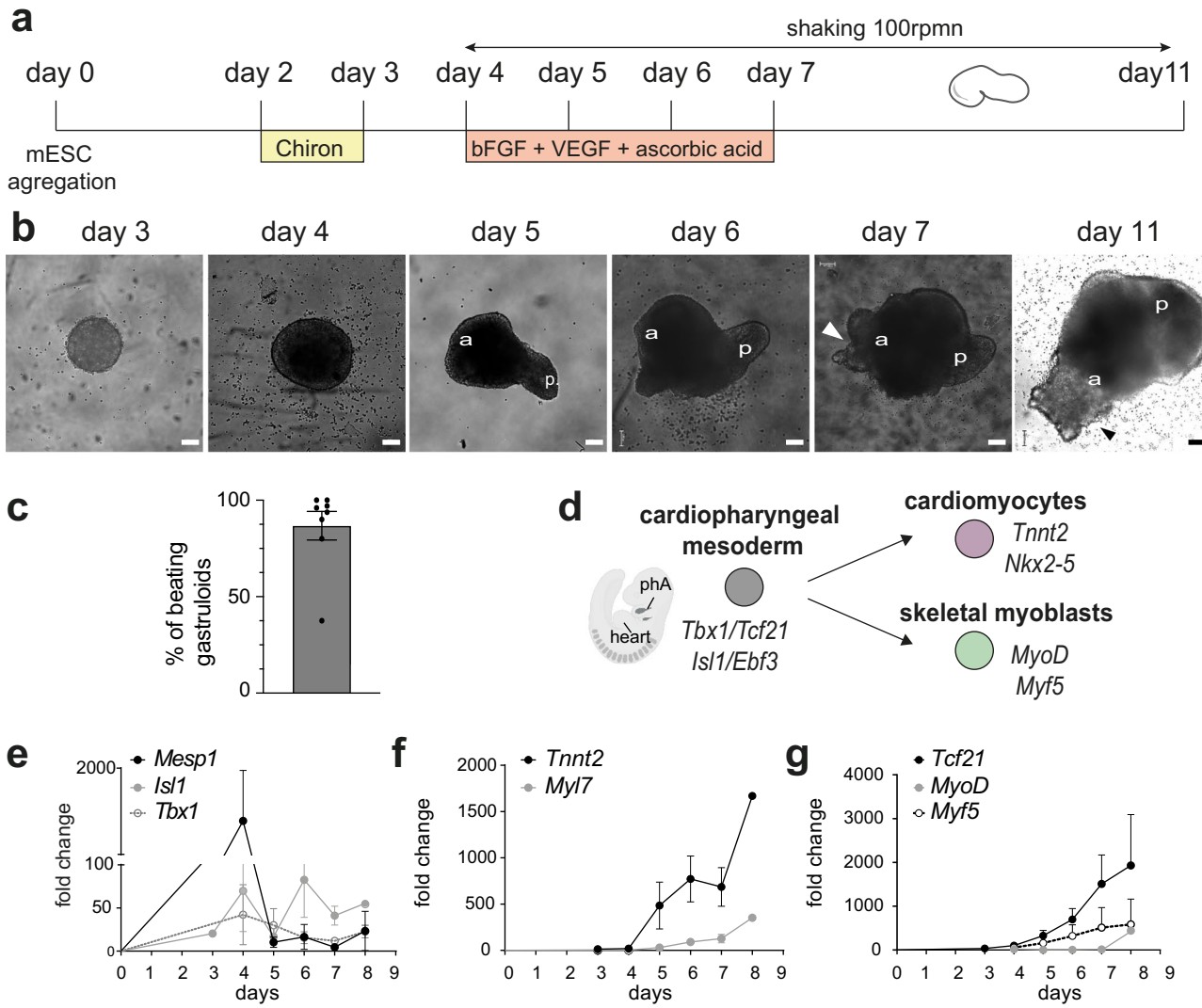

**Fig. 1 | The culture of gastruloids for 11 days allows the transcription of markers of the CPM and their derivatives. a** Experimental scheme of the generation of gastruloids from mouse embryonic stem cells (mESCs). Gastruloids were cultured in N2B27 up to 11 days, with the addition of small molecules from day 4 to day 7 as indicated below the time line. **b** Representative images of gastruloids at day 3, day 4, day 5, day 6, day 7 and day 11. Arrowheads indicate beating areas. **c** Percentage of beating gastruloids per experiments ($n = 8$ independent experiments). Data are presented as mean percentage ± SEM. **d** Scheme of CPM specification in mouse toward the myogenic fates. Created in BioRender. Lescroart, F. (2024) https://BioRender.com/m68f097. Markers of each state are indicated in italics. **e**–**g** Expression profiles of *Mesp1* ($n = 9$), *Isl1* ($n = 3$) and *Tbx1* ($n = 6$) (**e**), *Tnnt2* ($n = 9$), *Myh7* ($n = 3$) and *Myl7* ($n = 6$) (**f**) and *Tcf21* ($n = 8$), *Myod* ($n = 7$) and *Myf5* ($n = 5$ independent experiments) (**g**) throughout the culture of gastruloids as measured by quantitative RT-PCR. Results are normalized to the expression of *Tbp*. Fold changes are represented relative to expression on day 0. Data are presented as mean values ± SEM.

We then investigated the expression of additional markers of the CPM such as *Ebf3*, *Tcf21* and *Nkx2–5*, together with *Isl1* or *Tbx1*, at later time points in E7.75 to E9.5 embryos and day 5 to day 7 gastruloids (Fig. 2e–l and Supplementary Fig. 3c, d). In the embryo, *Nkx2–5* expression labeled the heart tube while *Isl1* and *Tbx1* were expressed in second heart field (SHF) progenitors, located in pharyngeal mesoderm behind the differentiated heart tube (Fig. 2e–j and Supplementary Fig. S3c). *Ebf3* and *Tcf21* are expressed more broadly, but they are also found in the pharyngeal region of the mouse embryo, where their expression overlaps with *Tbx1* and *Isl1* ((Fig. 2e–j). *Nkx2–5* was also expressed in the anterior part of the gastruloid, marking cardiomyocytes (Fig. 2g, h and Supplementary Fig. 3d). On days 5 and 6, gastruloids expressed *Tbx1* and *Isl1* in a domain adjacent to that of *Nkx2–5* (Fig. 2g, h and Supplementary Fig. 3d). In gastruloids, between day 5 and day 7, both *Tcf21* and *Ebf3* are expressed in overlapping domains with *Isl1* and *Tbx1* (Fig. 2g, h, k, l). In day 7 gastruloids, *Ebf3* and *Tcf21* were also found in a close domain of expression adjacent to the cardiomyocytes (*Tnnt2* +) (Supplementary

Fig. 3f). Similarly, in day 11 gastruloids, *Tbx1* were also found in a close domain of expression adjacent to *Tnnt2*+ cells (Fig. 2m, n). Among all the gastruloids analyzed for *Tnnt2* expression, 100% showed a positive expression domain at both day 7 ($n = 14$ across 5 independent experiments) and day 11 ($n = 23$ across 5 independent experiments) (Fig. 2o). Taken together, these findings indicate that CPM and cardiomyocyte markers are expressed in a similar spatio-temporal pattern in both gastruloids and mouse embryos.

To further investigate whether myoblast differentiation occurs and is faithfully recapitulated in gastruloids, we also compared the expression of myoblast markers in both models. In stage E9.5 mouse embryos, *Myf5* positive cells marked skeletal myoblasts, notably in the somites (Fig. 2p). Similarly, *Myf5* were detected close to *Tcf21* positive cells in gastruloids collected at day 11 (Fig. 2q). Detailed analysis revealed that 84.4% of the gastruloids analyzed showed *Myf5* expression ($n = 45$ across 6 independent experiments) (Fig. 2r). Similarly, we analyzed with HCR, the expression of *Myh3*, a marker of committed

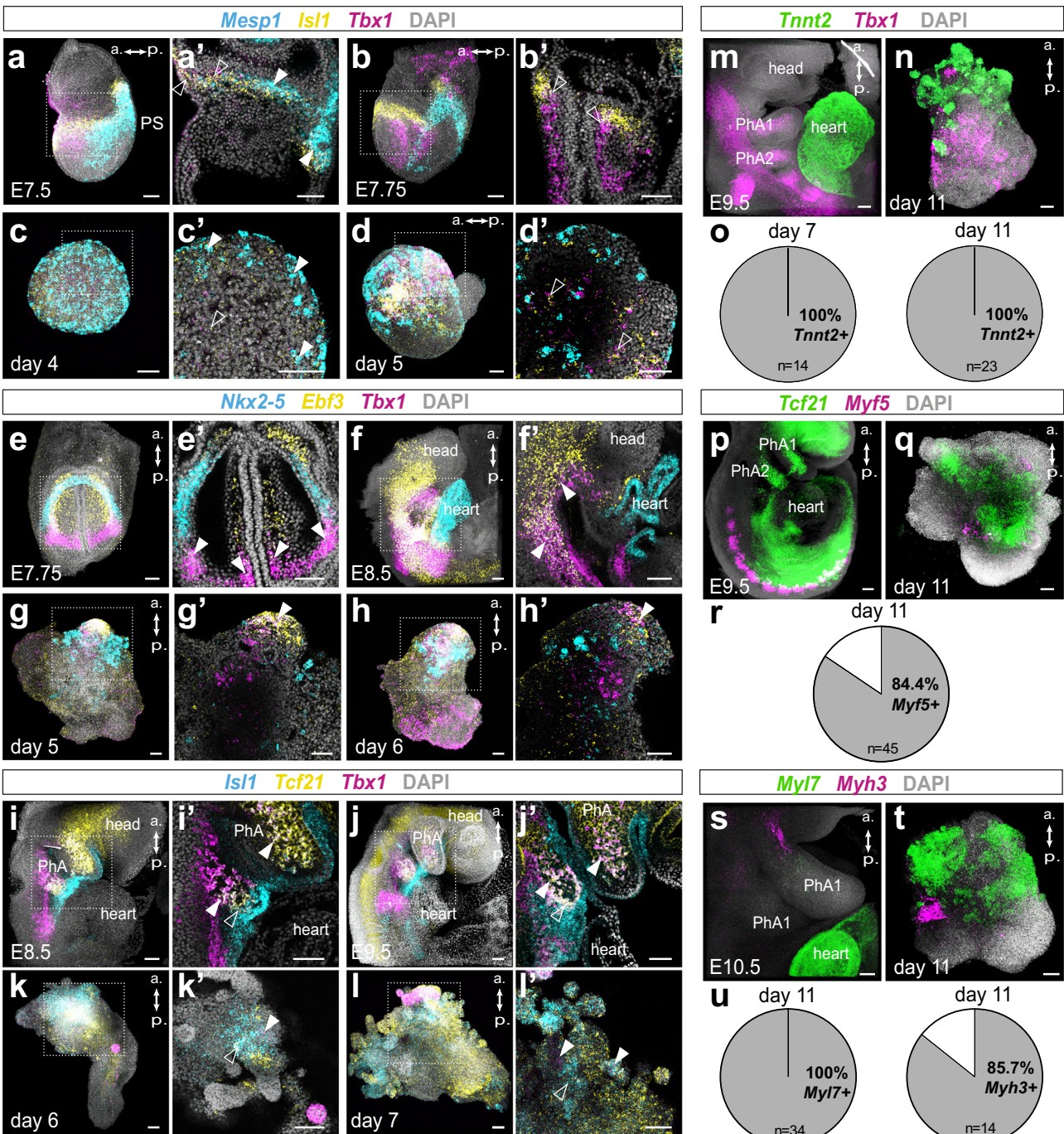

**Fig. 2 | Gastruloids faithfully recapitulate CPM specification and differentiation toward its myogenic fates. a–d** Representative maximum intensity projection (MIP) of mouse embryos at E7.5 (**a**), E7.75 (**b**), gastruloids on day 4 (**c**) and day 5 (**d**) after RNAscope with *Mesp1* (cyan), *Isl1* (yellow) and *Tbx1* (purple) probes. **a′**, **b′**, **c′**, and **d′**: Optical sections of the area indicated with dotted lines. Arrowheads indicate overlapping expression of *Mesp1* and *Isl1* (white) or of *Tbx1* and *Isl1* with no expression of *Mesp1* (empty). *n* = 4 embryos and *n* > 5 gastruloids.
**e–h** Representative MIP of embryos at E7.75 (**e**), E8.5 (**f**), gastruloids on day 5 (**g**) and day 6 (**h**) after RNAscope with *Nkx2–5* (cyan), *Ebf3* (yellow) and *Tbx1* (purple) probes. **e′**, **f′**, **g′**, and **h′**: Optical sections of the area indicated with dotted lines. White arrowheads indicate overlapping expression of *Ebf3* and *Tbx1*. *n* = 4 embryos and *n* > 5 gastruloids. **i–l** Embryos at E8.5 (**a**), E9.5 (**b**), gastruloids on day 6 (**c**), and day 7 (**d**) after RNAscope with *Isl1* (cyan), *Tcf21* (yellow) and *Tbx1* (purple). **i′**, **j′**, **k′**,

and **l′**: Optical sections of the area indicated with dotted lines. Arrowheads indicate overlapping expression of *Tbx1* and *Tcf21* (white) or of *Isl1* and *Tcf21* (white). *n* = 4 embryos and n > 5 gastruloids. **m, n** Representative MIP of the pharyngeal region of embryo at E9.5 (**m**) and gastruloid on day 11 (**n**) after RNAscope with *Tnnt2* (green) and *Tbx1* (purple). **o** Percentage of gastruloids expressing *Tnnt2* on day 7 (left - *n* = 14) and day 11 (right − *n* = 23). **p, q** Representative MIP of an embryo at E9.5 (**p**) and gastruloid on day 11 (**q**) after RNAscope with *Tcf21* (green) and *Myf5* (purple). **r** Percentage of gastruloids expressing *Myf5* on day 11 (*n* = 45). **s, t** Representative MIP of the pharyngeal region of an embryo at E10.5 (**s**) and gastruloid on day 11 (**t**) after HCR with *Myl7* (green) and *Myh3* (purple). **u** Percentage of gastruloids expressing *Myl7* on day 11 (left - *n* = 34) or expressing *Myh3* on day 11 (right − *n* = 14). a, anterior, p, posterior, PS, primitive streak, PhA, pharyngeal arch. Scale bars: 100 μm. All data derived from at least 2 independent experiments.

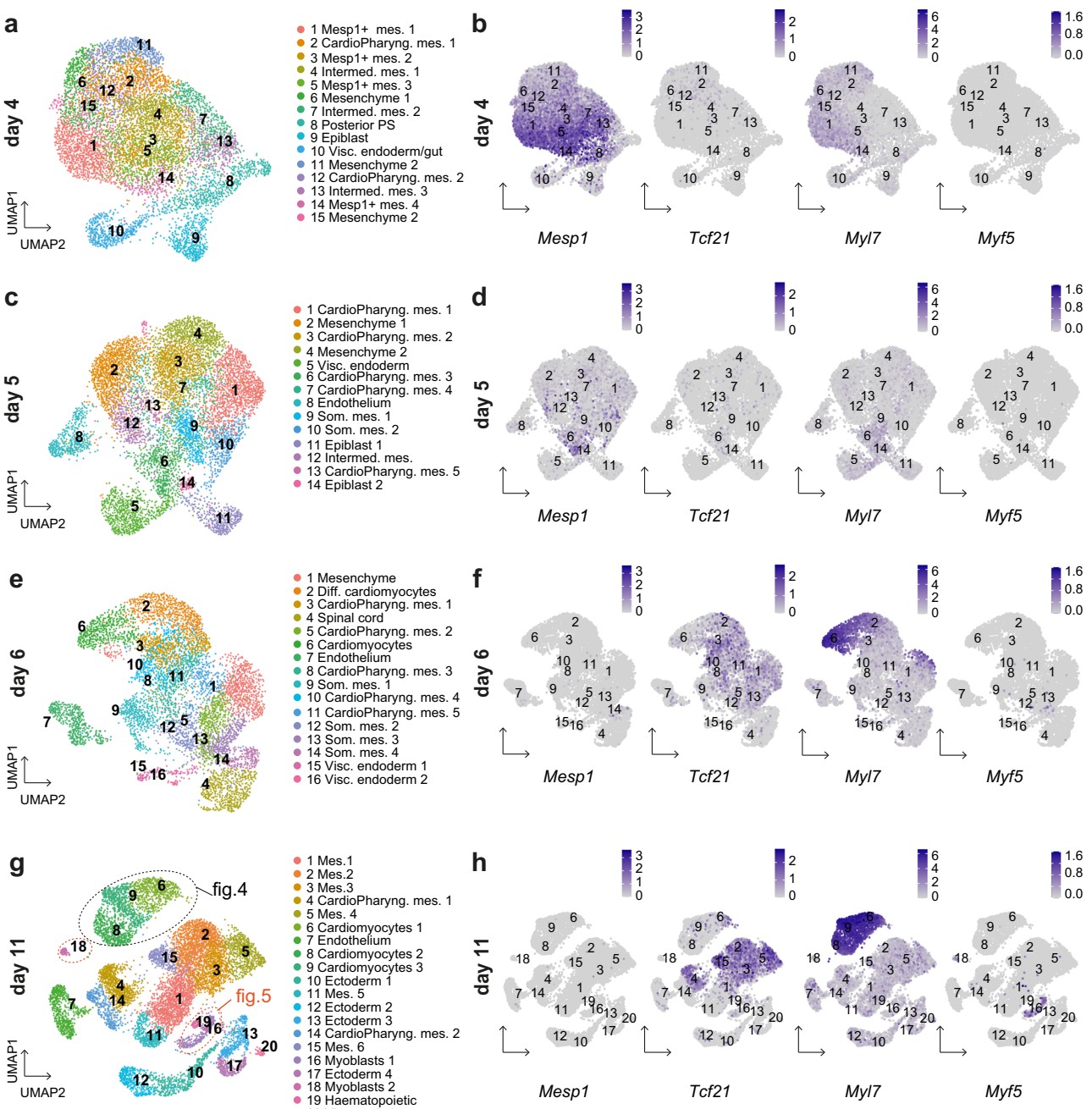

**Fig. 3 | Transcriptomic analysis reveals cardiopharyngeal mesoderm sub-populations in gastruloids.** UMAP representation of Leiden clustering of gastruloids at day 4 (**a**), day 5 (**c**), day 6 (**e**) and day 11 (**g**). FeaturePlots of key markers in gastruloids at day 4 (**b**), day 5 (**d**), day 6 (**f**) and day 11 (**h**). Scale bars represent expression levels. Mes., mesoderm; CardioPharyng., cardiopharyngeal; Intermed., intermediate; Visc., visceral; Som. Somitic; Diff., differentiating; PS, primitive streak.

skeletal myoblasts, together with *Myl7*, a marker of cardiomyocytes. At E10.5, *Myh3* is detected in the first pharyngeal arch (Fig. 2s). On day 11, gastruloids also showed *Myl7* and *Myh3* expression. *Myh3* expression domain was located slightly more posterior than *Myl7* expression domain (Fig. 2t). Similar to *Tnnt2*, *Myl7* expression was detected in all gastruloids analyzed at day 11 (*n* = 34 in >5 independent experiments) (Fig. 2u). *Myh3* was not detected in all gastruloids, but 85.7% of gastruloids analyzed showed co-expression of *Myl7* and *Myh3* (*n* = 14 in 4 independent experiments) (Fig. 2u). These data support the co-existence of skeletal myoblasts and cardiomyocytes in gastruloids cultured for over 7 days. Together, this comparison shows that gastruloids and mouse embryos display similar spatio-temporal gene expression during CPM development.

## Single-cell transcriptomic reveals CPM subpopulations in gastruloids

To further assess the potential of gastruloids to form CPM and investigate in detail the different cell populations, we performed single-cell RNA-sequencing (scRNA-seq) on gastruloids following a time course. Gastruloid cells were collected at day 4, day 5, day 6 and day 11. At each time point, at least 8 gastruloids were dissociated into single-cell suspensions, and approximately 7,000 cells per sample were sequenced. After quality controls, we analyzed 6,646 cells at day 4, 6,704 cells at day 5, 5,284 cells at day 6 and 8,024 cells at day 11. We performed Leiden clustering and differential gene expression analysis at each time point (Fig. 3 and Supplementary Data 1). Integration of the gastruloid single-cell data with the published atlas of mouse embryonic cells

**Table 1 | Percentage of positive cells for specific genes in scRNAseq**

|        | Tbx1  | Mesp1  | Tnnt2  | Myf5  |
|--------|-------|--------|--------|-------|
| Day 4  | 1.43% | 66.08% | 2.03%  | 0.03% |
| Day 5  | 2.55% | 17.77% | 9.02%  | 0.1%  |
| Day 6  | 1.97% | 1.06%  | 37.07% | 0.61% |
| Day 11 | 1.52% | 0.17%  | 35.22% | 1.41% |

ranging from E6.5 to E8.5[40] (see Methods 4–5) showed that cells collected at day 4, day 5 and day 6 likely overlap with cells of E7.25-E7.75, E8.0-E8.25 and E8.5 embryos respectively (Supplementary Fig. 4), in agreement with our fluorescent in situ hybridization experiments (Fig. 2a–l). However, gastruloid cells collected at day 11 do not overlap in the UMAP with cells of the atlas, likely reflecting distinct transcriptional profiles (Supplementary Fig. 4j, k). Gastruloid cells at day 11 might thus reflect more differentiated states, usually found after E8.5 in mouse embryos. This observation indicates that while our gastruloid single-cell dataset from day 4, day 5 and day 6 can be confidently compared with the cells of the atlas, distinct annotation criteria had to be applied to day 11 clusters. Cell type annotation transfer, from the atlas, was first applied to the primary clusters of day 4, 5 and 6 (See Methods 4). Manual annotation was then performed based on this label transfer and on differential gene expression (Supplementary Data 1). The day 11 clusters were annotated manually based on differential gene expression analysis (Supplementary Data 1) and reference to tissue database (https://tissues.jensenlab.org/About).

To assess the cellular heterogeneity of the gastruloids, we analyzed the different clusters obtained at each time point. At day 4, we observed a significant number of clusters of mesodermal cells (Fig. 3a, b; Supplementary Figs. 5 and 6) with four clusters showing high *Mesp1* expression (clusters 1, 3, 5 and 14) (Table 1). Analysis of the single-cell data also showed high expression of the *Gata4* and *Gata6* transcription factors in most clusters (all clusters except clusters 8, 9 and 13) (Supplementary Fig. 7). Gata transcription factors are known to have a broad expression pattern in the anterior/cardiogenic mesoderm as well as in the endoderm[41]. They have a critical function during heart development[42] and recent work in the fish has shown that they are involved in CPM specification, by promoting cardiac fate and inhibiting pharyngeal identity[43,44]. *Hand1*, a marker of the extra-embryonic mesoderm and cardiac progenitor cells in the recently identified juxtacardiac field[45,46] was also highly expressed at day 4 (except in clusters 8, 9 and 10) (Supplementary Fig. 7). Interestingly, with scRNA-seq, we detected *Myl7* as early as day 4 (Fig. 3b). Cluster 9 showed expression of the pluripotency marker *Pou5f1* indicative of epiblast-like cells (Supplementary Fig. 7). *Mesp2*, the closest homolog of *Mesp1*, was only expressed at very low levels in gastruloids (Supplementary Fig. 7).

At day 5, the cell type annotation transfer from the atlas was enriched in mesenchymal and pharyngeal mesodermal cells while nascent mesoderm was barely observed (Supplementary Fig. 8). We identified 5 clusters annotated as pharyngeal mesoderm (clusters 1, 3, 6, 7, and 13) (Fig. 3c). The first fully differentiated cells were found in cluster 8. This cluster corresponds to endothelial cells and is marked by the expression of *Sox7* (Supplementary Figs. S8 and S9). At this stage, *Mesp1* started to be downregulated while *Gata6* was still highly expressed (Fig. 3d and Supplementary Fig. 9). *Isl1* expression was also detected, while *Tbx1* was present at a very low level and in a small number of cells (Table 1 and Supplementary Fig. 9).

At day 6, we uncovered 16 clusters (Fig. 3e, Supplementary Fig. 10). We detected two clusters of differentiating or differentiated cardiomyocytes (clusters 6 and 2) expressing high level of *Myl7* and other cardiac myosin genes (Fig. 3f). All these data are consistent with the previous report on gastruloids[35]. We also found 5 clusters of

pharyngeal mesodermal cells (clusters 3, 5, 8, 10 and 11). Expression of the transcription factors involved in CPM development, including *Tbx1*, *Tcf21*, *Lhx2* and *Ebf3*, was also detected (Fig. 3f and Supplementary Fig. 11). These data reveal the emergence of CPM in differentiating gastruloids.

Finally, to investigate the potential differentiation of CPM derivatives in gastruloids, we analyzed scRNA-seq data at day 11. We identified 20 clusters (Fig. 3g). One cluster (cluster 20) likely corresponds to visceral endoderm tissue, with expression of *Epcam* (Supplementary Fig. 12). We also uncovered 4 different clusters (clusters 10, 12, 13 and 17) of ectodermal cells with enrichment of genes linked with neural derivatives, including a small number of *Sox10*+ neural crest-like cells in cluster 13 (Supplementary Data 1 and Supplementary Fig. 12). We identified a cluster of endothelial cells (cluster 7) and a cluster of blood/hematopoietic lineages (cluster 19). Clusters 1, 2, 3, 4, 5, 11 and 14 were annotated as mesoderm and included mesenchymal cells and undifferentiated progenitors (Fig. 3g, Supplementary Data 1). The presence of high expression levels of *Myl7* in clusters 6, 8 and 9 indicates that these clusters contain cardiomyocytes. Strikingly, we found 2 clusters (clusters 16 and 18) with expression of the myogenic regulatory genes *Myf5* (Fig. 3g, h and Supplementary Fig. 5). Together, our data reveal different CPM subpopulations in the gastruloid model and show that markers of cardiomyocytes and skeletal myoblasts are found in this model.

## Different subtypes of cardiomyocytes differentiate in gastruloids

To investigate whether the cardiomyocyte clusters at day 11 represent different cardiac subpopulations, we performed a detailed analysis of the clusters 6, 8, and 9, i.e., the three clusters containing cardiomyocytes. Among the genes differentially expressed in cluster 6, we found *Actg1*, *Cald1*, *Actb*, *Acta2*, *Vsnl1*, *Gucy1a1*, *Shox2* and *Ptn* (Fig. 4a, b and Supplementary Table 1). This cluster thus showed the expression of *Cald1*, *Acta2*, expressed in smooth-muscle cells and throughout the myocardium of the immature heart tube[47,48] as well as genes involved in sinus venosus/sinus atrial node development (*Vsn1*, *Shox2*)[49–51] (Fig. 4a, b). In cluster 9, *Nppa*, *Itga6*, *Myl7*, *Myh6* were among the genes differentially expressed (Fig. 4c, d and Supplementary Data 1). These genes are usually enriched in atrial cardiomyocytes. Cluster 8, on the other hand, was enriched in genes such as *Myl2*, *Myl3*, *Myh7*, *Mpped2*, *Pln* (Fig. 4e, f and Supplementary Data 1). These genes were previously identified as signatures of ventricular cardiomyocytes in different independent studies[52–54]. Our scRNA-seq analysis thus identified 3 cardiomyocyte clusters with distinct transcriptional signatures. We can further speculate that gastruloids contain ventricular, atrial and conductive-like cardiomyocytes.

To validate the presence of atrial and ventricular cardiomyocytes in gastruloids we performed in situ hybridization using the HCR method with specific ventricular and atrial probes. We first showed that at E10.5, *Myl2* is indeed restricted to ventricular cardiomyocytes, while *Myl7* is expressed in all cardiomyocytes but enriched in atrial cardiomyocytes (Fig. 4g). We then performed HCR experiments in wholemount gastruloids or followed by flow cytometry (see Methods). We found clearly distinct domains of *Myl2* and *Myl7* expression, each exhibiting a unique but seemingly random organization (n = 13/15 - Fig. 4h) or in two well organized domains (n = 2/15 - Fig. 4h'). We also found a significant shift of the cloud of cells, in the flow cytometry chart, indicative that a significant proportion of cells are *Myl7*-positive alone (corresponding to clusters 6, 8, and 9 of the sc-RNAseq data) or *Myl7*-positive in combination with *Myl2* (differentially expressed in cluster 8) in gastruloids at day 11 (Fig. 4i). This result suggests that *Myl7*+, *Myl2*+ ventricular-like cardiomyocytes and *Myl7*+,*Myl2*- non-ventricular cardiomyocytes are present in gastruloids. This result further validates the scRNAseq analysis and confirms the existence of different subpopulations of cardiomyocytes in gastruloids.

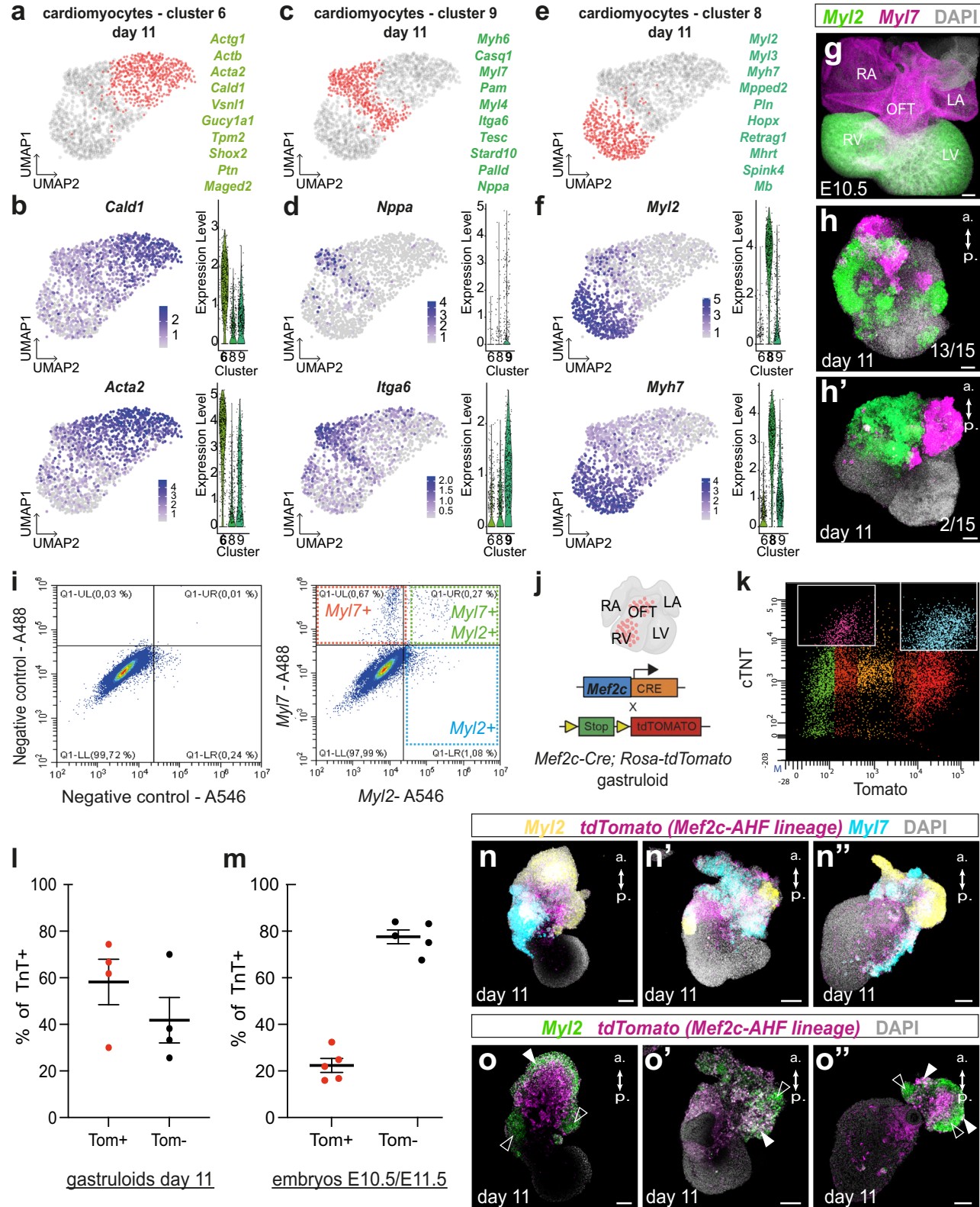

In order to support these results, we performed lineage tracing with a cell line, that labels specific subpopulations of cardiomyocytes in gastruloids. For that purpose, we rederived mESCs from *Mef2c-AHF-enhancer-Cre; Rosa^{tdTomato/+}* mouse blastocysts. *Mef2c-AHF-enhancer-Cre* mouse line is known to specifically label progenitors that contribute to outflow tract, right ventricle and a subpopulation of venous pole myocardium (dorsal mesenchymal protrusion) as well as non-cardiac

CPM derivatives[55–57] (Fig. 4j). *Mef2c-AHF-enhancer-Cre; Rosa^{tdTomato/+}* (*Mef2c-Cre; Rosa-tdTomato*) derived mESCs were able to form beating gastruloids (Supplementary Video 2) with expression of tdTomato in their anterior domain as expected. Flow cytometry at day 11 revealed that *Mef2c-Cre; Rosa-tdTomato*-derived gastruloids contain a significant proportion of cardiomyocytes, as shown by the expression of cardiac troponin T (cTNT) (Fig. 4k). Thus, tdTomato+ cardiomyocytes

**Fig. 4 | Different subtypes of cardiomyocytes differentiate in gastruloids.**
**a**, **c**, **e** UMAP representation of day 11 cardiomyocytes. Red dots represent cells of cluster 6 (**a**), cluster 9 (**c**) or cluster 8 (**e**). The most differentially expressed genes of the cluster are shown in green. **b**, **d**, **f** Feature and violin plots of key differentially expressed genes in the cardiomyocyte cluster 6 (**b**), cluster 9 (**d**) and cluster 8 (**f**) at day 11. **g** Representative maximum intensity projection image of a mouse embryonic heart at E10.5 after HCR experiment with *Myl2* (green) and *Myl7* (purple) probes. *n* = 2 hearts. **h**–**h′** Representative maximum intensity projection images of gastruloids at day 11 after HCR experiment with *Myl2* (green) and *Myl7* (purple). *n* = 15 gastruloids in 4 independent experiments. **i** Representative FACS analysis of the combined expression of *Myl7* and *Myl2* expression in all of the living gastruloids' cells at day 11 after HCR experiment. Negative controls with no probe are shown in left. *n* = 4 independent experiments. **j** Scheme of the experimental design with the use of the *Mef2c-AHF-enhaner-Cre; Rosa-tdTomato* (*Mef2c-Cre; Rosa-tdTomato*) line that label the right ventricle and outflow tract. Created in BioRender. Lescroart, F. (2024) https://BioRender.com/b93x660 **k.** Representative FACS analysis of the combined expression of tdTomato and cTnT expression in all of the living gastruloids' cells at day 11. **l**, **m** Graphs showing the proportion of tdTomato+ (Tom + ) cells in cardiomyocytes (cTnT+ cells) derived from day 11 gastruloids (**l**) and embryonic cells at E10.5/E11.5 (**m**) (in *n* = 4 and *n* = 5 independent experiment (mean with standard error of mean (SEM)). **n**–**n″** Confocal images (maximum intensity projection) of 3 independent *Mef2c-Cre; Rosa-tdTomato* gastruloids at day 11 after HCR with *Myl2* (yellow) *tdTomato* (purple) and *Myl7* (blue) probes. *n* = 16 gastruloids in 3 independent experiments. **o**–**o″** Optical sections from the same gastruloids shown in (**n**–**n″**) to specifically observe the co-expression of *Myl2* (green) and *tdTomato* (purple) marked by white arrowheads. Empty arrowheads indicate *Myl2+ tdTomato-* cells. Scale bars: 100 μm. LA, left atrium; RA, right atrium; LV, left ventricle; OFT, outflow tract; a, anterior; p, posterior.

represented an average of 58.2% (ranging from 30 to 74.4%) of all cardiomyocytes in gastruloids (Fig. 4l). Flow cytometry in E10.5/E11.5 mouse embryos showed less variability with between 0.9 and 1.6% of tdTomato+ cells. In addition, staining of mouse embryos with cTnT antibody showed that *Mef2c-AHF-enhancer-Cre* derived (tdTomato + ) cardiomyocytes accounted for about 22.9% of all cardiomyocytes in embryos (Fig. 4m). These data indicate that gastruloids can undergo differentiation into at least two distinct cardiomyocyte fates: either a tdTomato positive SHF fate (outflow tract, right ventricle for example) or a first heart field (FHF) enriched tdTomato negative fate.

We additionally performed in situ hybridization, with the ventricular specific *Myl2* probe, to discriminate left ventricular versus right ventricular fates in *Mef2c-AHF-enhancer-Cre; Rosa*[tdTomato/+] gastruloids collected at day 11 (Fig. 4n). We observed the presence of both *Myl2 +/ tdTomato+* and *Myl2 +/tdTomato-* domains (Fig. 4o), supporting the specification of right and left ventricular cardiomyocytes respectively. *Myl2-/Myl7 +/tdTomato+* cells also present in gastruloids might correspond to outflow tract-like cardiomyocytes. Interestingly, we also observed the expression of *Foxc2* and *Hoxb1*, markers of the anterior and posterior SHF respectively[58], in gastruloids collected at day 4 (Supplementary Fig. 13). In day 6 and day 11 gastruloids, we identified *Tbx5 +/Tnnt2+* double positive cells, as potential FHF-derived cardiomyocytes and *Tbx5 +/Tnnt2-* cells as potential posterior SHF progenitors[57] (Supplementary Fig. 13). Together these data validate the existence of different subpopulations of cardiomyocytes (left ventricular, right ventricular and atrial) in gastruloids.

## Skeletal myogenesis in gastruloids
To ascertain whether gastruloids could generate skeletal muscle cells, we analyzed the clusters 16 and 18, which are characterized by the expression of genes associated with myoblast differentiation (Fig. 5a). Differential gene expression between clusters 16 and 18 was not conclusive (Supplementary Data 2). We first analyzed the cell cycle and uncover that the myoblast cluster 16 was more enriched in cells in G2/M and S phases than cluster 18 (Fig. 5b). This suggests that cluster 16 might correspond to a more proliferative state than cluster 18. We thus decided to investigate in detail the genes involved in cardiopharyngeal and/or somitic mesoderm muscle differentiation and their expression in the two myoblast clusters (Fig. 5c). We noticed an increased expression of genes associated with muscle precursors such as *Pax7* and *Myf5* in cluster 16 compared to cluster 18, as shown by the violin plots (Fig. 5c, d). Although *Met* has been shown to be downregulated during differentiation in some particular CPM- or somites-derived myoblasts[59,60], it was expressed in more cells and at higher levels in cluster 16 compared to cluster 18 (Fig. 5d). In contrast, markers of more committed myoblasts such as *MyoD*, *Myog* and *Myh3* were enriched in cluster 18 (Fig. 5d). These data, together with our findings on the cell cycle, indicate that myogenesis occurs in the gastruloid model and that clusters 16 and 18 express markers of muscle precursors and committed myoblasts, respectively.

To dissect the CPM versus somitic origin of these myoblasts, we investigated the expression of transcription factors that were specific to either the CPM or somitic muscle progenitors (Fig. 5c). Expression of the CPM genes, *Isl1*, *Pitx2*, *Tbx1* and *MyoR*, was found in both cluster 16 and 18 but only in a small number of cells (Fig. 5e). *Tcf21* was expressed only in the myoblast cluster 16 in a limited number of cells. *Pax3* expression was also recorded in about 40 out of 243 cells of cluster 16 and in 2 out of 67 cells of cluster 18 (Fig. 5e). Deeper analysis of cluster 16 showed a mutually exclusive expression of *Tbx1* and *Pax3* in the population of muscle precursors, however a significant number of *Tbx1+* or *Pax3+* cells co-express *Myf5* and/or *Myod* (Fig. 5f, g). This observation suggests the activation of distinct myogenic programs. Our scRNA-seq analysis at day 11 thus indicates that both CPM and somitic progenitors undergo myogenesis in the gastruloid model.

To validate this data and further support the existence of myogenesis in gastruloids, we performed in situ hybridization with skeletal myocyte specific probes either in wholemount or followed by flow cytometry. Flow cytometry interestingly showed a shift in the cloud of cells with two cell populations expressing *Myog* (Fig. 5h). As found in the single-cell data, we identified a subpopulation of cells expressing *Myog* alone (cluster 16) and a subpopulation of cells expressing *Myog* and *Myh3* (cluster 18). Using immunofluorescence, we showed the expression of MyoG at the protein level, further supporting the specification and differentiation of skeletal myoblasts (Fig. 5i). Wholemount RNAscope experiments revealed *Myf5* expression near the *Pax3* and *Tcf21* expression domains (Fig. 5j). Furthermore, wholemount RNAscope experiments with *Pax3*, *Tbx1* and *Myf5* probes showed the expression of *Tbx1* and *Pax3* in distinct domains (Fig. 5k) and co-expression of *Myf5* with either *Pax3* (Fig. 5k') or *Tbx1* (Fig. 5k″), further supporting the scRNAseq dataset. Overall, these observations demonstrate the existence of convergent CPM and somitic myogenic programs in the gastruloid model.

## Muscular trajectories are found in gastruloids over time
Time series scRNA-seq data allows the reconstruction of transcriptional trajectories from a progenitor cell state toward differentiated cell states[61]. We hypothesize that we could similarly infer the transcriptional trajectories from CPM progenitors toward cardiomyocyte and myoblast cell states. We merged the single-cell data from the 4 different time points to create a transcriptional time series of the growing gastruloids (Fig. 6a). We first used CellRank (see Methods)[62]. We focused on a differentiated cell state (called macrostate), that includes cells from the cardiomyocyte cluster 8 from the day 11 dataset, and computed the fate probabilities towards this macrostate. Interestingly, we found high fate probability with cells from clusters 6 and 9 from day 11, as well with clusters annotated as cardiomyocytes at day 6 (Fig. 6b). We then focused on a second macrostate that includes cells from the myoblast cluster 18 from the day 11 dataset. We found high fate probabilities with cells from the myoblast cluster 16 also from

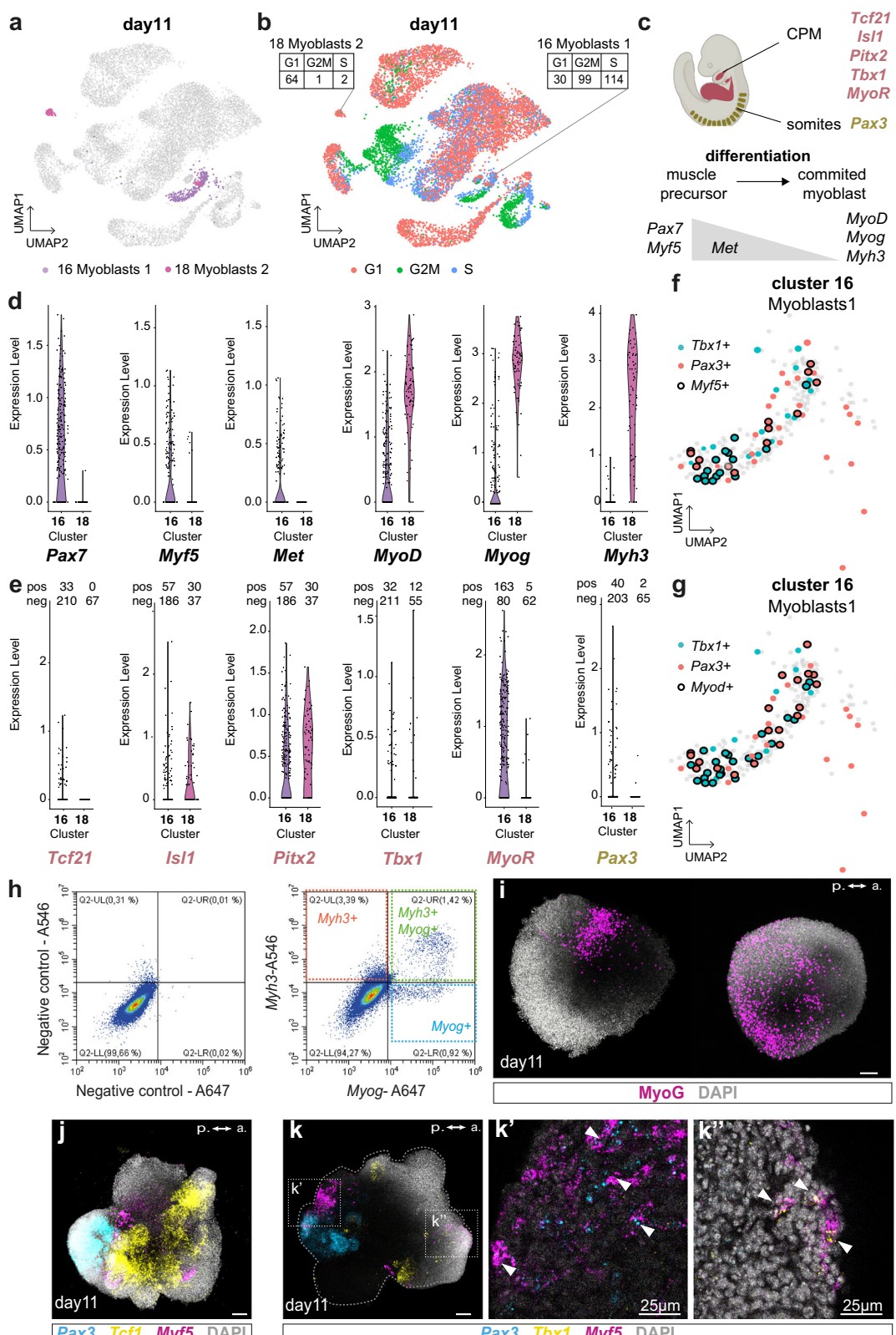

day 11 (Fig. 6c). These data indicate the existence of muscular trajectories toward the cardiac and skeletal muscle states.

To reconstruct fate decision trees for the various populations, we used URD[63] (see Methods) as a computational reconstruction method, with the merge of all time points (days 4, 5, 6 and 11). URD analysis computed transcriptional trajectories from cells at day 4 (named root) towards the different clusters at day 11 (named tips) (Fig. 6d). We

observed 5 branches emerging from day 4. The first branch (branch A) heads towards the ectodermal lineages (clusters 10, 12, 13 and 17) while the last branch (E) heads towards the endoderm lineages (cluster 20). The 3 other branches (branches B, C and D) are mesodermal, and include the endothelial (cluster 7) and hematopoietic lineages (cluster 19). Notably, one of these mesodermal branches, branch D, heads towards the cardiomyocytes (clusters 6, 8 and 9) and myoblast clusters

**Fig. 5 | Skeletal myogenesis takes place in gastruloids from the CPM and somitic mesoderm. a** UMAP representation of day 11 myoblast clusters highlighted in purple (cluster 16) and in pink (cluster 18). **b** UMAP representation of day 11 cells with their cell cycle status, G1 (red), G2/M (green) and S (blue). The numbers of cells in each phase of the cell cycle are indicated for the myoblasts cluster 16 and 18. **c** Scheme of the genetic regulation of skeletal myogenesis from the CPM (red) or somitic mesoderm (green). Created in BioRender. Lescroart, F. (2024) https://BioRender.com/o57x120. **d** Violin plots showing the level of expression of *Pax7*, *Myf5*, *Met*, *MyoD*, *Myog* and *Myh3* specifically in the myoblast clusters of gastruloids at day 11. **e** Violin plots showing the level of expression of *Tcf21*, *Isl1*, *Pitx2*, *Tbx1*, *MyoR* and *Pax3* specifically in the myoblast clusters at day 11. **f**, **g** UMAP representation of day 11 cluster 16. Blue and red dots represent *Tbx1* and *Pax3* expressing cells, respectively. Black circles represent *Myf5* (**f**) or *Myod* (**g**) expressing cells. **h** Representative FACS analysis of the combined expression of *Myh3* and *Myog*

expression in gastruloid cells at day 11 after HCR experiment. Negative controls with no probe are shown on the left. *n* = 4 independent experiments. **i** Representative maximum intensity projection (MIP) images of two gastruloids at day 11 after immuno-fluorescent experiment with MyoG antibody (purple). *n* = 13 gastruloids in 3 independent experiments. a, anterior; p, posterior. Scale bars: 100 µm. **j** Representative MIP of a gastruloid at day 11 after RNAscope experiment with *Tcf21* (yellow), *Pax3* (blue) and *Myf5* (purple). *n* = 10 gastruloids in 3 independent experiments. Scale bar: 100 µm. **k** Representative MIP of a gastruloid at day 11 after RNAscope experiment with *Tbx1* (yellow), *Pax3* (blue) and *Myf5* (purple). *n* = 9 gastruloids in 3 independent experiments. Scale bar: 100 µm. Gray dotted lines delinate the gastruloid. Optical sections of the area indicated with dotted lines are shown in (**k'–k''**). White arrowheads indicate overlapping expression of *Myf5* with *Pax3* (**k'**) or with *Tbx1* (**k''**). Scale bars for **k'** and **k''**: 25 µm. a, anterior; p, posterior.

(clusters 16 and 18). Thus, the URD analysis shows that transcriptional trajectories can be found toward both the cardiomyocytes and skeletal muscles.

To further dissect these trajectories, we specifically focused on branch D. This branch also includes clusters 4 and 14. These two clusters were previously defined as mesodermal derivatives (Fig. 3g). We investigated gene expression in branches M and H to characterize further these clusters (Fig. 6d). We found that branches M and H showed *Gata6* expression as well as *Hand1* and *Mab21l2*, which mark the juxta-cardiac field[45,46] (Supplementary Fig. 14). We also showed expression of *Tbx18* and *Wt1* (Supplementary Fig. 14), which mark the proepicardial organ and their derivatives[64]. It is tempting to speculates that these clusters contain epicardial cells but none of these markers are completely restricted to this cell lineage.

We then investigated how key markers of the CPM and their derivatives are expressed in the tree. We found that *Isl1*, *Tcf21*, but also *Lhx2* and *Ebf3* are expressed in branch D, in the root as well as along the trajectories towards the cardiomyocyte and myoblast clusters (Fig. 6e, f and Supplementary Fig. 14). *Myl7* is specifically expressed in the three cardiomyocyte clusters (6, 8 and 9) and the cardiomyocyte branching point I (Fig. 6g), while *Myl2* is expressed after the branching bifurcation only in cluster 8 (Fig. 6h). As expected *Myog* is expressed in branch L that heads towards the myoblast cluster 18 (Fig. 6i). *Tbx1*, *MyoR*, markers of the CPM muscle progenitors, *Pax3*, markers of the somitic muscle progenitors, and the early myogenic transcription factor *Myf5* are all expressed in branch F that heads towards the two myoblast clusters 16 and 18 (Supplementary Fig. 14). We also explored where cells of clusters from days 5 and 6 would be found in this tree (Fig. 6j–n). Strikingly, cells of the pharyngeal mesoderm 1 cluster, identified at day 5, were found almost at the root of the tree (branch D), before the branching bifurcation between the cardiomyocytes and myoblast lineages (Fig. 6j).

Similarly, cells of the cardiopharyngeal mesoderm 2 cluster, identified at day 6, were found before the branching point (branch D) (Fig. 6k). However, we found that cells of the cardiopharyngeal mesoderm 3 cluster, identified at day 6, were restricted to the cardiomyocyte lineages (branch I). These cells are located after the branching bifurcation between the cardiomyocytes and myoblast lineages (branch D) (Fig. 6l). We then focused on the two myoblast clusters identified at day 11. Cells of cluster 16 were also found in branch F, before the branching point that heads toward the myoblast clusters 16 and 18 (Fig. 6m). Cells of cluster 18, on the other hands, were found only after branch bifurcation, in branch L (Fig. 6n). These data further confirm that myoblast cells from cluster 18 are in a more differentiated state that cells from cluster 16. Together these trajectory inferences show that muscular transcriptional trajectories are found in gastruloids. They also show that a specific branch of the tree includes the CPM and heads towards both cardiomyocytes and myoblasts clusters.

## Discussion

Our results provide evidence that CPM markers are expressed in a similar spatio-temporal pattern in gastruloids and mouse embryos. We also showed that CPM differentiates in the gastruloid model into different subpopulations of cardiomyocytes as well as skeletal myoblasts. Gastruloids thus faithfully recapitulate the developmental timing of CPM specification and differentiation (Fig. 7).

Here we provide an in vitro model for parallel differentiation of cardiomyocytes and skeletal muscles. Indeed, in addition to cardiomyocyte differentiation previously reported by Rossi and colleagues[35], our study demonstrates that skeletal myogenesis occurs in gastruloids. Using scRNA-seq analysis we found that the mesodermal origin (CPM or somitic) was not reflected in the clustering of skeletal muscle progenitor cells. In contrast, we identified two clusters that segregate different states of myogenesis. This finding indicates that transcriptional similarity due to the state of myogenesis might predominate over its developmental origin, raising challenges in addressing whether myogenesis in gastruloids derives from CPM or somitic mesoderm.

We first showed the expression of key markers of CPM progenitors such as *Tcf21*, *Tbx1*, *Lhx2*, *MyoR*, and *Pitx2*[16,20,24,25]. Expression of transcription factors, such as *Tbx1*, is however relatively low in our scRNA-seq analysis (Table 1). This could be explained by the low sequencing saturation (see Methods), since RNAscope experiments have confirmed the expression of such key CPM markers in gastruloids. Additionally, we identified transcriptional trajectories from the CPM towards both myoblasts and cardiomyocytes. These trajectories should be interpreted with caution, as the time intervals between time points may be too large to assure accuracy. However, we independently observed the expression of myogenic transcription factors either in close proximity to or co-expressed with *Tcf21* or *Tbx1* in gastruloids, using wholemount RNAscope and/or HCR. These results indicate that skeletal myogenesis, at least in part, occurs from the CPM. Based on the exclusive expression of *Tbx1* (CPM marker) and *Pax3* (somitic marker) in myoblast cluster 16 we propose that myoblasts also derive from somitic mesoderm and that gastruloids thus allow skeletal myogenesis of both head and trunk programs. This has been further demonstrated with the co-expression of *Myf5* and *Pax3* in gastruloids by RNAscope.

Furthermore, transcriptomic trajectories computed with URD showed a branch with both cardiomyocytes and myoblasts. But expression of both somitic (*Pax3*) and CPM markers is found in that particular branch. This highlights the known limitations with the use of single-cell transcriptomics approaches for lineage inference. ScRNA-seq alone provides information only on gene expression, and attempts to reconstruct lineage trajectories from transcriptomic data can be misleading and do not allow a complete lineage reconstruction[65]. Only clonal analysis will confirm whether CPM bipotent progenitors are found in gastruloids but this requires specific tools and analysis.

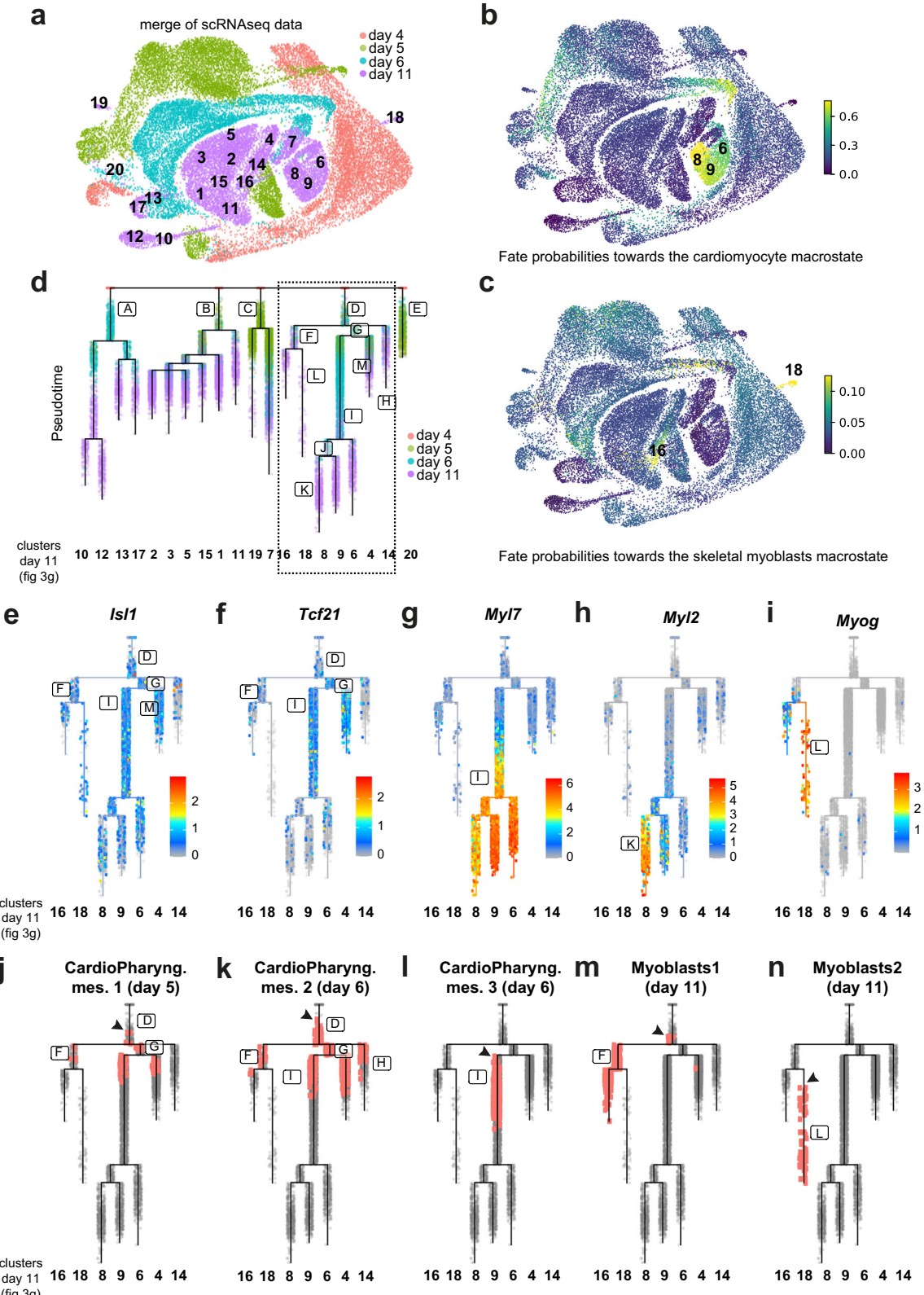

Skeletal myogenesis remains limited in gastruloids, with cluster 16 and 18 comprising only about 4% of all cells (Table 2). Further work is needed to determined how to better promote skeletal myogenesis through the addition of growth factors (HGF, IGF, FGF) or inhibitors (BMP or Rock-inhibitors)[29,30]. Similarly, in the context of CPM development, it will be important to establish methods to robustly promote CPM at the expense of more posterior/somitic mesoderm. This may be

achieved by reducing or inhibiting Wnt and/or Nodal activity, as shown previously in 2D culture[30].

Interestingly, different subpopulations of cardiomyocytes were identified in gastruloids. Gene expression analysis suggests that these subpopulations correspond to ventricular (*Myl2+*, *Myl3+*, *Myh7+*), atrial (*Myh6+*, *Nppa+*) and immature/conductive cardiomyocytes *(Shox2+, Vsnl1+)*[49–52,54]. However, atrial and ventricular

**Fig. 6 | Myogenic trajectories are found in gastruloids over time. a** UMAP representation of all gastruloid cells merged from day 4 (red), day 5 (green), day 6 (blue) and day 11 (purple). Numbers represent identified clusters of day 11 (as defined in Fig. 3g). **b, c** Fate probabilities towards the cardiomyocyte macrostate (**b**) or towards the skeletal myoblast macrostate (**c**). Scale bars represent absorption probabilities. Numbers represent key cardiomyocytes and myoblast clusters of day 11. **d** URD trajectories showing transition from cells at day 4 (root) towards cells at day 11 (tips). Framed letters represent branches along the tree. The dotted box represents the lineage branch detailed in (**f**–**o**). **e**–**i** Expression of *Isl1* (**e**), *Tcf21* (**f**),

*Myl7* (**g**), *Myl2* (**h**) and *Myog* (**i**) across URD trajectory. Scale bars represent expression levels. **j**–**n** Representation of clusters' cells across URD trajectory. Red dots represent cells of the cardiopharyngeal mesoderm (CardioPharyng. mes.) cluster 1 of day 5 (in **j**, Fig. 3c, cluster 1), of the cardiopharyngeal mesoderm (CardioPharyng. mes.) cluster 2 of day 6 (in **k**, Fig. 3e, cluster 5), of the cardiopharyngeal mesoderm (CardioPharyng. mes.) cluster 3 of day 6 (in **l**, Fig. 3e, cluster 8), of the myoblast cluster 1 of day 11 (in **m**, Fig. 3g, cluster 16) and of the myoblast cluster 2 of day 11 (in **n**, Fig. 3g, cluster 18).

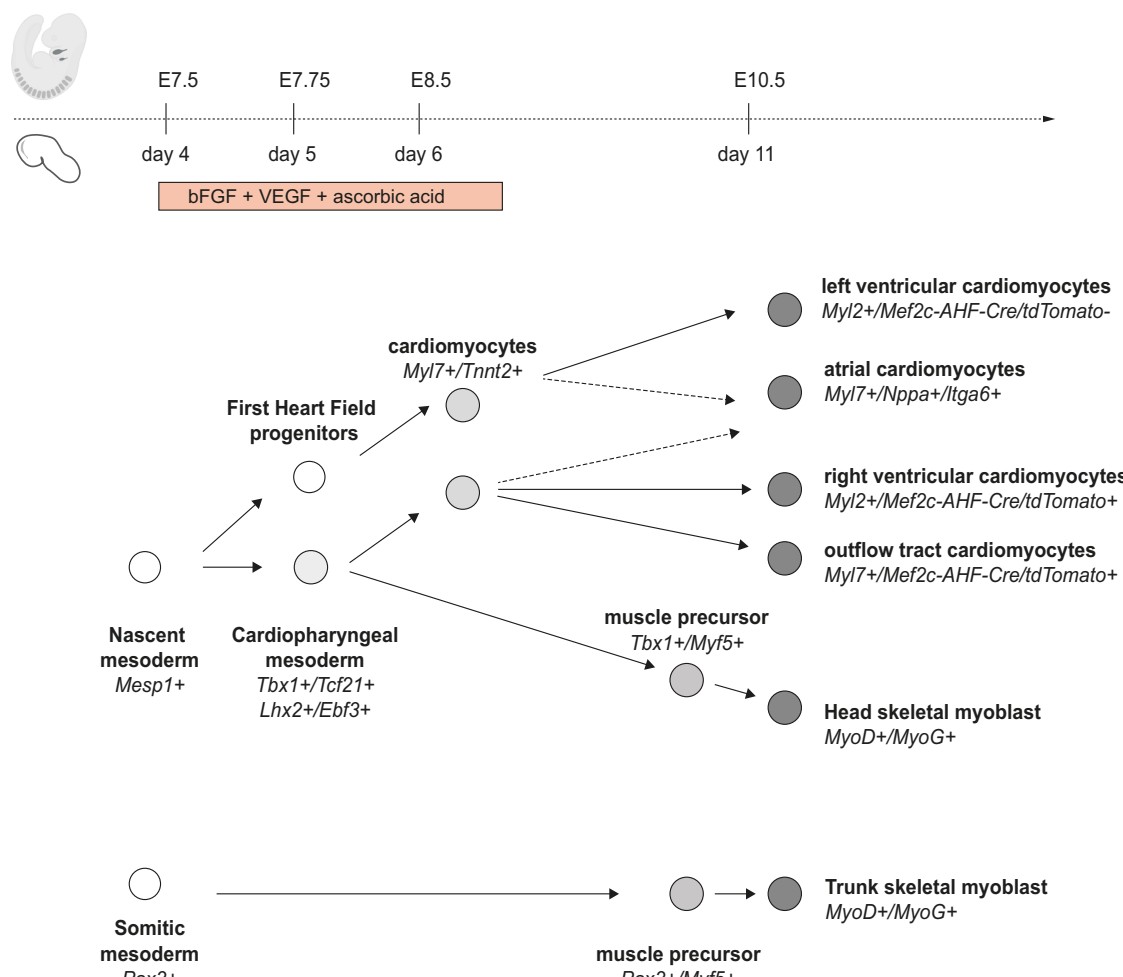

**Fig. 7 | Schematic representation of the developmental timing of CPM specification and differentiation in gastruloids in comparison with the mouse model.** The scheme of the embryo has been created in BioRender. Lescroart, F. (2024) https://BioRender.com/m68f097.

myocardium share many genes in common during development. They are distinguished later by acquisition of their mature electro-mechanical properties (conduction, voltage tension, some structural proteins and ions channels). Thus, few genes are differentially expressed in vivo: *Myl7* is only restricted to atrial myocardium in fetal mouse embryo, while *Myl2* is restricted in ventricular myocardium. Finally, *Nppa* is detected in the atrium and the ventricle

during development, but its expression becomes atrial specific immediately before birth[66]. It is thus not clear how mature are the cardiomyocytes in gastruloids. We confirmed however the restricted expression of *Myl2* in ventricular cardiomyocytes in E10.5 mouse hearts. We showed with multiplex fluorescent in situ hybridization that we could find different domains of *Myl2* and/or *Myl7* expression in gastruloids further supporting the idea that gastruloids allow the specification of atrial and ventricular cardiomyocytes.

Using the *Mef2c-AHF-Cre/tdTomato* mESC line, we also found that left ventricular (*tdTomato-/Myl2+*) and right ventricular cardiomyocytes can differentiate in day 11 gastruloids. Additionnally, *tdTomato+/Myl7+/Myl2-* cells observed in gastruloids may correspond to outflow tract cardiomyocytes. This aligns with previous observations that markers of the FHF, as well as the anterior and posterior SHF, are expressed in gastruloids[35].

**Table 2 | Number of cells per day 11 clusters in scRNAseq**

| Clusters | 1 | 2 | 3 | 4 | 5 | 6 | 7 | 8 | 9 | 10 |
|---|---|---|---|---|---|---|---|---|---|---|
| Nbr. of cells | 1052 | 934 | 658 | 583 | 538 | 499 | 456 | 444 | 427 | 385 |
| **Clusters** | **11** | **12** | **13** | **14** | **15** | **16** | **17** | **18** | **19** | **20** |
| Nbr. of cells | 340 | 296 | 291 | 279 | 257 | 243 | 185 | 67 | 56 | 34 |

Further work still will be needed to improve morphogenesis and the spatial organization of these different subpopulations. It will be also important to play with small molecules inhibitors/activators and/or growth factors to increase or reduce specifically some cardiomyocytes subpopulations. For example, it is conceivable that adding retinoic acid could enhance posterior SHF specification and promote the differentiation of atrial cardiomyocytes[53,67,68]. Further work will be also required to determined how other cardiac cell types (endocardium, proepicardial organ and its derivatives) develop in gastruloids.

Our results reveal robust myogenic differentiation to heart and head muscle in the gastruloid system. This is supported by reproducibility with similar observations from different replicates. We also showed redundancy with similar conclusions obtained when using scRNA-seq, RNAscope, HCR, lineage tracing and immunofluorescence. The use of different mESC lines showed the robustness of the gastruloids for the specification of the CPM. However, our gastruloid model also presents some limitations. In contrast to gastruloids from human induced pluripotent stem cells, which have been shown to form a heart-tube like structure[34] or human cardioids which have been shown to form multi-chamber heart like structures[69], our murine gastruloid model does not show robust heart morphogenesis. We also have not observed cardiac crescent like structures, as described previously[35]. This indicates that further studies and protocol adaptations are required to form these different chamber structures and to improve cardiac morphogenesis. Interactions between pharyngeal endoderm and the heart are known to be crucial for heart morphogenesis[70,71]. We hypothesize that with the robust development of CPM in our study, the spatial proximity between the cardiac derivatives and the endoderm/gut structure might be perturbed. Immunofluorescence experiments with E-cadherin antibody at day 7 and day 11 suggest indeed that endodermal derivatives are present but may be reduced compared to what has been previously reported[35]. Affecting the proximity with the endoderm might impact cardiac morphogenesis.

Morphogenesis and physical forces are also different in gastruloids, compared to embryos. Our results indicate that while CPM is formed in gastruloids, there is no indication of pharyngeal arch morphogenesis. Further studies are required to determine how these transient structures are formed in the embryo. The gastruloid model offers a possibility to reconstruct such processes and investigate whether physical constraints could play a role as shown for example with matrigel embedding for somitogenesis[36,37]. Additionally, we know that pharyngeal arches are also colonized by neural crest cells, which are mostly absent in the gastruloid model[33]. Adapting the protocol to support both neural crest cell and CPM development might improve pharyngeal arch morphogenesis. While cranial/cardiac neural crest cells are known to be critical for outflow tract and valve development[72–74] as well as for the patterning of head branchiomeric skeletal muscles[75], our results confirm that neural crest cells are not required for cardiac and CPM specification.

As previously reported, we have also noted variability among the lines of mESC used. Notably, mESC derived from *Mef2c-AHF-Cre/tdTomato* blastocysts seem to have less potential for skeletal myogenesis. We speculate that this could be due to a different genetic background. Alternatively, established and rederived mESC lines from mouse embryos might not strictly have the same pluripotency status. In particular, we found that different mESC clones exhibited a different beat and myogenic potential. Due to the lack of specific markers, it was also difficult to address whether other CPM derivatives such as connective tissue or smooth muscle[56,59,76], could develop in gastruloids.

Several reports have already indicated that gastruloids, somitoids, segmentoids and embryonic trunk-like structures, from mouse or human pluripotent stem cells, could faithfully recapitulate paraxial mesoderm early specification with the formation of somites[36,37,77–79]. None of these studies however showed the initiation of myogenesis.

However, most of their protocols stopped after 5 to 7 days of culture whereas we have extended the culture of gastruloids up to 11 days. Our kinetics of expression showed that markers of myogenesis are initiated only after day 8. This suggests that somitoids, segmentoids and embryonic trunk-like structures might also have the potential to undergo myogenesis if cultured for a longer period of time.

Together our results show the relevance of the gastruloid model to investigate CPM specification and differentiation, and demonstrate that this in vitro model could be used as an alternative model to complement or reduce the use of animal models. The protocol is relatively simple and allows the culture of a large number of gastruloids in a single experiment. It thus offers a powerful in vitro platform to model congenital heart disease and muscle dystrophies (22q11.2DS for example), and to deconstruct CPM specification as well as myogenesis. CPM development is still not fully understood in vivo. The field lacks reporter lines and faces challenges in carefully tracking the progenitors and their derivatives along a course of about 3 to 5 days, from the emergence of the progenitors at gastrulation to their differentiation into cardiomyocytes or skeletal myoblasts. CPM-containing gastruloids additionally offer the possibility for high resolution live imaging and cell tracking. Furthermore, the model will likely be useful for large scale genetic and drug screens to better understand and potentially identify therapeutic strategies for congenital diseases affecting head and heart muscles. In this context, we expect gastruloids to become indispensable tools in the field of cardiac and skeletal muscle specification to address the mechanisms of lineage specification as early as gastrulation.

## Methods

### Mice colonies

*Mef2c−AHF-Cre* and *Rosa-tdTomato* (*Gt(ROSA)26Sor^tm9(CAG-tdTomato)Hze*) mice (mixed background) were previously described[55,80]. We used CD1/Swiss mice (Charles River) as wildtype animals. Adult breeding mice (*Mus musculus*) in our colony range 2–8 months age. We did not genotype for sex of the embryos. Mice were housed at a temperature between 20 and 22 °C with a level of humidity between 40 and 60%. Each cage was provided with food, water and two types of nesting materials. Mice colonies were maintained in a certified animal facility with 7 h to 19 h light cycle. Mouse colonies were maintained in certified animal facilities (agreement #C 1301308) in accordance with European guidelines. The experiments were approved by the local ethical committee (Aix-Marseille Univ CEBEA CE14 directed by Erica Lopez) and the study is compliant with all relevant ethical regulations regarding animal research (Ministère de l'Education Nationale, de l'Enseignement Supérieur et de la Recherche; Authorization N 32-08102012).

### Cell culture

R1/E #1036 mouse embryonic ES cells were purchased from ECACC (cat. Number 07072001). Zx1 mouse ESCs were obtained from M. Kyba[81]. *Mef2c-cre Rosa tdTomato* cells were derived in the lab from E3.5 bastula collected from mouse[55]. All cell lines were maintained on 0.1% gelatin coated labware in GMEM-BHK21 medium (Gibco ref 21710-025) supplemented with 8% Fetal Bovine Serum (Pansera, ref P30-2602), 1% non-essential amino acids (Gibco 11140-035), 1% Sodium Pyruvate (Gibco ref 11360-039), 1× Penicillin/Streptomycin (Gibco ref 15070-063), 0.1 mM beta-mercaptoethanol (Sigma ref M3148-100ML). This basal medium was completed with 5 ng/ml recombinant mouse LIF (Sigma ref LIF2050), 3 μM CHIR99021 (Sigma ref SML1046-5MG) and 1 μM PD0325901 (Sigma ref PZ0162-5MG). Cells were passaged at least twice but no more than 10 times before differentiation.

### Gastruloid production

Gastruloids were produced according to previously described protocols[33,35] with slight modifications (centrifugation for aggregation and culture in N2B27 medium after day7). Briefly, mES reaching a 60%

confluency were dissociated with 0,5% Trypsin-EDTA (Gibco ref 25300-054). After enzyme neutralization, cells were carefully washed 2 times with D-PBS (Gibco ref 14190-094) and resuspended in N2B27 medium (Neurobasal Medium (Gibco ref 21103049), DMEM/F12 (Gibco ref 11320074), 0,5% B27 supplement (Gibco ref 17504-044), 0,5% N2 supplement (Gibco ref 17502-048), 1× Glutamax (Gibco ref 35050-038), 0.1 mM beta-mercaptoethanol (Sigma ref M3148-100ML). In total 300–600 individual cells were distributed into 40 μl of N2B27 in each well of a Ultra-Low-Attachment 96 well round bottom plate (Corning ref 7007). Plates were then centrifuged at 100 g for 5 min and placed in a 5% $CO_2$ incubator during 48 h. Then, 150 μl of N2B27 with 1–3 μM CHIR99021 were added in each well and plates were incubated for 24 h. After this period, medium was replaced by N2B27 without CHIR99021 for another 24 h. Then from day 4 to day 7, N2B27 was supplemented with 30 ng/ml bFGF (R&D ref 233-FB), 5 ng/ml VEGF (Gibco ref PHC9394) and 0,5 mM Ascorbic Acid (Sigma ref 255564-5 G) and culture plates were placed on an orbital shaker (N-BIOTEK NB-1015RC) at 100 rpm. After day 8, N2B27 was used and changed every other day. As previously reported[32,35], different cell lines respond differently to Chir (CHIR99021) concentration. Depending on the stocks of Chiron we had to adjust a bit the concentration of Chiron. *Mef2c-cre; Rosa-tdTomato* line elongates better with 1 μM of Chir.

### Gene expression analysis by QPCR

Day 0 undifferentiated mES cells and different time points gastruloids were collected during the experiments. Samples were then lysed in Buffer RA1 and RNA was extracted with NucleoSpin RNA kit (Macherey-Nagel ref 740902.250). Total RNAs were then retrotranscribed to cDNA using AffinityScript Multiple Temperature cDNA Synthesis Kit (Agilent ref 200436) and QPCR were run on QuantStudio 5 according to PowerUp™ SYBR® Green instructions (Thermo ref A25742). List of QPCR primers can be found in supplementary information (Supplementary Table 2). Data were all standardized using *Tbp* as a housekeeping reference gene and on day 0 results and gene expression fold changes were calculated with the ΔΔCt method.

### Single cell RNA-sequencing analysis

Gastruloids collected on day 4, 5, 6, and 11 were dissociated by incubation 10 min at 37 °C in 0.5% Trypsin-EDTA. Cells were washed 2 times with D-MEM + 10% FBS and filtrated through a 40 μm cell strainer. Cell count and viability were determined on a Countess II (Thermo ref AMQAX100). Single cell-RNA seq libraries were made from 10,000 individual cells using 10× Genomics Chromium Next GEM Single Cell 3′ kit v3.1 (PN-1000268) and the Chromium Controller. Libraries were then sequenced on an Illumina NextSeq 500 at the MMG Sequencing platform.

For each time point, we obtained:

day 4: 57 682 mean reads/cell, 4 684 mean genes/cell and 21.2% sequencing saturation.

day 5: 61 601 mean reads/cell, 4 944 mean genes/cell and 28,7% sequencing saturation.

day 6: 71 931 mean reads/cell, 4 782 mean genes/cell and 40.8% sequencing saturation.

day 11: 39 931 mean reads/cell, 3 282 mean genes/cell and 40.1% sequencing saturation.

Single-cell analysis was performed using (1) Seurat, the R toolkit for single-cell genomics[82], (2) CellRank, to discover the cellular dynamics[62], and (3) URD to obtain transcriptional trajectories[63]. The analysis was implemented in R v4.1.3 and Python v3.9.10.

### Preparation of the Pijuan-Sala et al. atlas

The Pijuan-Sala et al. atlas has been constructed from mouse embryos to describe gastrulation and early organogenesis[40]. It contains single-cell analysis at multiple developmental stages between 6.5 and 8.5 days post-fertilization[40]. The atlas was subsetted to select the cells with more than 200 feature gene counts and the features detected in at least 4 cells. Then, normalization was performed on the feature counts with the "LogNormalize" method. An atlas must be comparable to the dataset under study. Here, and because we work on gastruloids mimicking gastrulation, the stage "mixed_gastrulation" and the cells annotated "endoderm" or "extraembryonic endoderm" were removed from the atlas. Finally, the identification of the highly variable genes, the data scaling, and the PC analysis and UMAP dimensional reduction were performed as detailed in step 2 of the next section.

### Processing of the four individual single-cell datasets

Four single-cell RNA sequencing datasets were generated from gastruloids'cells collected at days 4, 5, 6, and 11. Each dataset was analyzed with the R package Seurat v4.1.1 using the following six steps[82].

### Quality controls

Low quality cells were filtered out with different cutoffs on gene expression. All these cutoffs were set manually and independently on each dataset. The proportion of transcripts mapping to mitochondrial and ribosomal genes were evaluated. We set cutoffs on high mitochondrial and low ribosomal gene expression. In addition, we set two cutoffs on the UMI counts of each cell. Finally, the data were subsetted to keep cells with more than 200 feature gene counts and features detected in at least 4 cells.

### Preprocessing workflow

Feature counts were normalized using the "LogNormalize" method. Highly variable genes were identified with the "vst" selection mode, and selected the 2000 most variable genes. Then, the data were scaled on all the features and Principal Component (PC) analysis dimension reduction was performed. The 30 first PCs were kept for the rest of the analysis. To visualize the data, Uniform Manifold Approximation and Projection (UMAP) was applied[83]. Finally, the normalized data were structured with a shared nearest neighbor graph. The number of nearest neighbors was set to 20 (default). We used the Leiden algorithm with the resolution set to 1 to cluster the graph[84].

### Doublets detection

Doublets detection was carried out with the R package DoubletFinder v2.0.3[85]. A cell type annotation transfer was first applied. Indeed, according to DoubletFinder recommendations, a cell type annotation transfer allows a better estimate of the number of detectable doublets[85]. The Seurat tutorial "Mapping and annotating query datasets" was applied to transfer the annotation of cell types, using the atlas of Pijuan-Sala et al. as a reference [40,82]. We estimated the number of heterotypic doublets from the percentage of doublets of the used reagent kit of 10x Genomics Chromium 3′ chemistry v3.1. In our case, the rate of doublets amounts to 8%. For each dataset independently, the pK parameter was optimized with the function paramSweep_v3. The first 30 PCs were considered. As a result of the process, every cell was labeled as either singlet or doublet. We selected the cells labeled as singlets, and reapplied the Step 2 preprocessing workflow to the singlet subset.

### Cell type annotation transfer from the atlas

Like for doublets detection, cell type annotations were transferred using the Seurat functions with default parameters[82].

### Global structure-preserving UMAP plots

The global structure-preserving UMAP dimension reduction was set up using the R package uwot v0.1.14[83]. The aim was to keep a similar vizualisation structure among the different datasets. The reference global structure was created using the Pijuan-Sala et al. atlas[40]. We first

integrated our dataset with the Pijuan-Sala et al. atlas using the Seurat framework for datasets integration with default parameters.

Besides, UMAP dimension reduction was carried out on a subset of the atlas containing 200 cells per cell type. From the resulting UMAP coordinates, the centroid of each cell type was calculated. We randomly assigned positions to all the cells of the integrated object (dataset and atlas) around the centroid of their corresponding cell type. The UMAP algorithm was run on the integrated object, with the random positions around cell type centroids as the starting points. In a following step, the UMAP algorithm was run again on the dataset only, with the dataset's UMAP coordinates in the integrated object as the starting points.

### Differential expression analysis

Differentially expressed genes (DEGs) were determined using a Wilcoxon Rank Sum test with the Seurat function FindAllMarkers. The function compares each cluster (group 1) to the rest of the data (group 2). Default values were used for min.pct, logfc.threshold, and return.thresh parameters: min.pct allows to speed up the function by selecting features that are expressed over a given proportion of cells within each group, return.thresh selects p-values lower than a given threshold, and logfc.threshold selects features showing at least a 2-fold change between the two groups of cells. We selected the positive markers only.

For each cluster, we also retrieved the most represented cell type and its corresponding percentage of cells in the cluster (see Table S1).

### Processing of the merged single-cell datasets

The four single-cell datasets were also merged. The datasets were reduced to the previously identified high-quality cells (see steps 1 and 3 of the processing pipeline). The identification of the highly variable genes, the data scaling, and the PCA and UMAP dimensional reductions were performed as described in step 2 from the previous section. Steps 4 to 6 were then applied to the merged dataset.

### Differential expression analysis for cardiomyocytes and myoblasts

Additional differential expression (DE) analyses were performed on two subsets of the day11 dataset. First, to better characterize the cardiomyocytes, the clusters identified as cardiomyocytes (ie. clusters 6, 8, and 9) were compared. In this case, each cluster was compared to the two others. The cluster 18 of day 11 was also identified as a cluster of cardiomyocytes (see Table S1). However, the day 11 is poorly annotated because the reference atlas does not reach this time points. We manually assigned the "myoblast" label to this cluster. DE analysis was performed between the clusters 16 and 18, two clusters of myoblast cells. In both analyses, the parameters used for the DE analyses were the same as detailed in step 6 of the previous section.

### Data preparation for file format conversion

CellRank and URD necessitate different file format. We started from raw data and removed the cells and features that were filtered out during the preprocessing analysis of the four datasets (step 1 and 3). The obtained datasets were then merged and the feature counts were normalized using the "LogNormalize" method. The object was saved as a.rds file (R).

### Cell Rank time series framework

In order to use CellRank[62] v1.5.1, the Seurat[82] object (R) was converted to an AnnData[86] v0.7.8 object (Python). The prepared dataset was converted into.h5ad (Python) (see Data preparation for file format conversion section).

The.h5ad object was preprocessed with loading, normalization, logarithmization, highly variable genes identification, PC analysis, and KNN graph computing, using the Scanpy v1.9.1 tool[87]. We added the metadata of UMAP coordinates, cell type assignation, and Leiden clustering to each cell. CellRank was used with the Waddington Optimal Transport method (WOT), the time series flavor of CellRank[88]. The WOT kernel was then computed. Due to insufficient proliferation genes in highly variable genes, we were not able to compute the initial growth rates. Hence, according to CellRank specifications, the transition matrix was computed with the parameter growth_iters = 3[62]. A symmetric transition matrix was also computed based on transcriptomic similarity, called connectivity kernel. WOT kernel and connectivity kernel both contributed to the estimator with the respective proportions of 0.8 and 0.2. The estimator was initialized with the Generalized Perron Cluster Cluster Analysis (GPCCA) method[89]. Based on Schur decomposition, 6 macrostates were computed[90]. Then, fate probabilities were computed towards the macrostates with the function compute_absorption_probabilities, with default parameters. Putative driver genes were identified with the function compute_lineage_drivers.

### Trajectory inference

The R package URD v1.1.1 was used for trajectory inference[63]. The prepared dataset was converted into an urd object (see Data preparation for file format conversion section). The variable features within each dataset were identified with the findVariableGenes function. A KNN graph was constructed with distances between cells' gene expression, considering the 60 nearest neighbors of each cell. We removed the outliers, identified visually according to the distance between the first and the 60th neighbors of each cell. 92 outlier cells were removed from the analysis. Then, a diffusion map was computed with the parameters sigma = 3 and knn = 60. The pseudotime was simulated 5 times starting from any cell of the day 4 and the results were averaged. All the clusters of day11 were defined as tips. The function pseudotime Determine Logistic was used with the parameters optimal.cells.forward = 40 and optimal.cells.backward = 80 to bias the input transition matrix of the random walk simulations. In total 10000 random walk simulations were carried out. Finally, the tree was constructed by merging the tips as soon they share a common path among all the random walk simulations. To build the tree, the parameters of the function buildTree were set as follows: bins.per.pseudotime.window=8, cells.per.pseudotime.bin = 28, divergence.method = 'preference', p.thresh = 0.001.

### RNAscope experiments on sections

Gastruloids collected at different time points were washed once in PBS + 0.01% Tween 20 (PBST) and then incubated in 4% PFA overnight at 4 °C. After 3 washes in PBST, approximately 8 Gastruloids were embedded in heat liquified Histogel (Epredia HG-4000-012). Solidified blocks were then post-fixed with 4% PFA during 1 h at room temperature and dehydrated with successive incubations of 30 min in Ethanol/PBST: 70%, 96%, twice 100%. Xylene was then added twice for 15 min at room temperature. Histogel blocs were finally embedded in heated liquid Paraplast plus (Sigma ref P3683-1KG). Tissue sections cut at 7 μm were then processed according to the protocol of the RNAscope Multiplex Fluorescent v2 kit (ACD-Bio cat. no.323110). The following probes were used: *mm-Ebf3-C3* (576871-C3), *mm-Isl1-C2* (cat no. 451931-C2), *mm-Mesp1-C2* (cat no. 436281-C2), *mm-Mesp1-C3* (cat no. 436281-C3), *mm-Myf5-C1* (cat no. 492911), *mm-Myod1-C2* (cat no. 316081-C2), *mm-Nkx2-5-C2* (cat no. 428241-C2), *mm-Tbx1-C1* (cat no. 481919), *mm-Tcf21-C2* (cat no. 508668-C2). Sections were imaged using an LSM800 confocal microscope (Zeiss) with the Zen software v3.9 and v3.10.

### Wholemount RNAscope experiments

Gastruloids and embryos were washed once in PBS + 0.01% Tween 20 (PBST) and then incubated in 4% PFA overnight at 4 °C. Gastruloids and embryos were then dehydrated in progressive bath in Methanol for storage. After rehydratation, RNAscope was performed according to

the protocol of the RNAscope Multiplex Fluorescent v2 kit (ACD-Bio cat. no.323110). Probes were incubated overnight at 40 °C. The following probes were used: *mm-Ebf3-C3* (576871-C3), *mm-Isl1-C2* (cat no. 451931-C2), *mm-Mesp1-C2* (cat no. 436281-C2), *mm-Mesp1-C3* (cat no. 436281-C3), *mm-Myf5-C1* (cat no. 492911), *mm-Myod1-C2* (cat no. 316081-C2), *mm-Nkx2-5-C2* (cat no. 428241-C2), *mm-Tbx1-C1* (cat no. 481919), *mm-Tcf21-C2* (cat no. 508668-C2), *mm-Hoxc4-C1* (cat no. 487941), *mm-Tnnt2-C4* (cat no. 418681-C4), *mm-Pax3-C4* (cat no. 455801-C4). To develop probes, we used Opal dyes from Akoya Bioscience (Opal-520, 1:250, Opal-570, 1:500 and Opal-650; 1:1000). Mounting was performed between two coverslips with RapiClear 1.49 (SunJin lab RC149001). Gastruloids and embryos were imaged using an LSM800 confocal microscope (Zeiss) with the Zen software v3.9 and v3.10. Maximum intensity projections were generated on Zen software v3.9 and v3.10 or on ImarisViewer v10.1.0.

### Wholemount HCR experiments

Gastruloids and embryos were washed once in PBS + 0.01% Tween 20 (PBST) and then incubated in 4% PFA overnight at 4 °C. Gastruloids and embryos were then dehydrated in progressive bath in Methanol for storage. HCR was performed according to Molecular Instruments protocols. For the permeabilization, gastruloids and embryos younger than E9.5 were incubated 15 minutes in a solution of protease K (10 μg/ml) at room temperature. Embryos at E9.5 and 10.5 hearts were incubated 20 min, while E10.5 embryos were incubated 25 min. The following HCR probes were used: *tdTomato-B5*, *Myl2-B2*, *Myl7-B3*, *Myh3-B2* and *Myog*-B1 with the following amplifier: B1-Alexa647, B2-Alexa488, B2-Alexa546, B3-Alexa488, B3-Alexa647 and B5- Alexa 546. Mounting was performed between two coverslips with RapiClear 1.49 (SunJin lab RC149001). Gastruloids and embryos were imaged using an LSM800 confocal microscope (Zeiss) with the Zen software v3.9 and v3.10. Maximum intensity projections were generated on Zen software v3.9 and v3.10 or on ImarisViewer v10.1.0.

### Wholemount immunofluorescence

Gastruloids were incubated in 4% PFA for 30 min at 4 °C and then washed in PBS + 0,01% Tween 20 (PBST). We incubated gastruloid in blocking solution for 2 hours at room temperature (5% horse serum, 1% bovine serum albumin in PBS + 0,01% Tween 20). Primary antibodies (mouse anti-cTnT (1:100, Invitrogen MA5-12960, clone 13–11), goat anti-VEGFR2 (1:100, R&D AF644), rat anti-E-cadherin (1:100, Takara M108, clone eccd-2), mouse anti-MyoG (1:10, DSHB, F5D-S)) were incubated for 2 days in the blocking solution. After several washes, secondary antibodies (donkey anti-mouse-IgG Alexa555 (1:400, Invitrogen A-31570), donkey anti-goat-IgG Alexa647 (1:400, Invitrogen A-21447), donkey anti-rat-IgG Alexa488 (1:400, Invitrogen A-21208), goat anti-mouse-IgG Alexa647 (1:400, Invitrogen A-12235)) were incubated for 2 days in the blocking solution. Mounting was performed between two coverslips with RapiClear 1.49 (SunJin lab RC149001). Gastruloids were imaged using an LSM800 confocal microscope (Zeiss) with the Zen software v3.9 and v3.10. Maximum intensity projections were generated on Zen software v3.9 and v3.10 or on ImarisViewer v10.1.0.

### Flow cytometry

Gastruloids collected on day 11 were dissociated by incubation 10 min at 37 °C in 0,5% Trypsin-EDTA. Cells were washed 2 times with D-MEM + 10% FBS and filtrated through a 40 μ cell strainer (Falcon 352340). After a PBS wash, 1 million cells were incubated for 20 min at room temperature with Zombie viability dye (Biolegend, 77143). Then cells were incubated in PBS + 10% FBS to block Fc receptors, followed by a permeabilization/fixation with Cytofix/Cytoperm BD kit (ref 554714). After washing, anti-cTnt antibody (Invitrogen MA5-12960) was incubated at a 1/400 dilution in BD wash buffer during 20 min at room temperature. Then cells were again washed and followed by an incubation with anti-mouse IgG coupled to PE-Cy7 (Biolegend, 406613) at a 1/200 dilution in BD wash buffer during 20 min at room temperature. After three washes, cells were resuspended in PBS + 1% BSA (Sigma ref A8412-100ML), filtered on a 40 μ cell strainer and analyzed on a BD FACS Aria II cytometer and analyzed with FACS Diva v9.0.1 (BD Biosciences).

Gastruloids made with *Mef2c-cre;Rosa-tdTomato* mESC were prepared by the same procedure up to Zombie viability dye labeling and directly analyzed on FACS Aria II to allow sorting of the Tomato positive and negative cells. RNA from sorted cells were extracted with NucleoSpin RNA extraction kit.

### HCR RNA flow analysis

Gastruloids collected on day 11 were dissociated by incubation 10 min at 37 °C in 0.5% Trypsin-EDTA. Cells were washed 2 times with PBS + 10% FBS and filtrated through a 40 μ cell. After PBS wash, 1 million cells were incubated for 20 min at room temperature with Zombie viability dye. Cells were then fixed by incubation with 4% PFA (EMS15714) during 1 h at room temperature. HCR multiplexed RNA detection was realized according to Molecular Instruments protocols. Cells were washed in PBST: PBS + 0,1% Tween 20, resuspended in 70% Ethanol and kept overnight at 4 °C. The day after, cells were washed twice in PBST and incubated at 37 °C in hybridization buffer. Probes specific for the different genes were then added at 16 nM concentration and incubated overnight at 37 °C. Cells were then washed 5 times with wash buffer and 2 times with 5× SSC (Gibco 15557-044) + 0.1% Tween 20. Then incubation of the cells with amplification buffer was done during 30 min at room temperature. Amplification hairpin h1 and h2 were prepared separately by heating at 95 °C for 90 s and cooling to room temperature for 30 min. Then hairpins were pooled at 60 nM and incubated with cells overnight at room temperature. After 6 washes with 5× SSC + 0.1% Tween 20. Samples were filtered on Flowmi 70 μm (Bel-Art H136800070) and analyzed on a Beckman Coulter Cytoflex LX (AMUTICYT platform) with CytExpert v2.5. The following HCR probes were used: *Myl2-B2*, *Myl7-B3*, *Myh3*-B2 and *Myog*-B1 with the following amplifier: B1-Alexa647, B2-Alexa546 and B3-Alexa488.

### Reporting summary

Further information on research design is available in the Nature Portfolio Reporting Summary linked to this article.

## Data availability

The scRNA-seq data generated in this study have been deposited in the Gene Expression Omnibus (GEO) database under accession code GSE232773. The processed single-cell data are available at https://doi.org/10.5281/zenodo.14188076. The qPCR and flow cytometry generated in this study are provided in the Supplementary Information/Source Data file. Source data are provided with this paper.

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

## Acknowledgements

We want to thank M. Kyba for providing the Zx1 mES cell line. Single-cell isolation, sequencing and alignment were performed with Veracyte or in the MMG GBIM facility. We thank the cells and animal imaging platform and the animal phenotyping core platform (MMG). Flow Cytometry was performed on the Amu-Flow Cytometry platform (Manon Richaud) and AMUTICYT platform (Stéphane Robert). We thank Lionel Spinelli and the CIML bioinformatics platform for expert advice on bioinformatics ana-lyses. We thank Luna Di Nardo for technical help with some RNAscope experiment. We thank Robert Kelly and Anabela Bensimon-Brito for careful reading of the manuscript. F.L.'s laboratory was supported by the INSERM ATIP-Avenir program. C. Choquet has been supported by the Fondation Lefoulon Delalande and N. Nandkishore by the Fondation

pour la Recherche Médicale (FRM). A.B. and C. Chevalier received funding from the "Association Française contre les Myopathies" [MoThARD-Project]. Computations were run on the Core Cluster of the Institut Français de Bioinformatique (IFB) (ANR-11-INBS-0013). F.L. and S.Z.'s laboratories were supported by the Agence Nationale pour la Recherche (ANR-Heartbox, ANR-Heartbound and ANR-Heartist) and the "Association Française contre les Myopathies" [MoThARD-Project]. Financial support from Inserm through the Booster Program Mecacell3D (F.L. and S. Z.)

## Author contributions

L.A., F.L., S.Z. designed the experiments. L.A. and C. Choquet performed most of the biological experiments. C. Chevalier performed bioinformatic analysis for all the single-cell RNA-seq data. L.A. performed HCR and FACS experiments. F.L. and C. Choquet performed RNAscope on gastruloids and mouse embryos. N.N. performed immunofluorescence and experiments with the Mef2c-Cre/tdTomato line. L.A. and N.N. rederived Mef2c-Cre/tdTomato mouse ESCs. A.G. provided technical support. F.L., S.Z. and A.B. wrote the manuscript. All authors read and approved the final manuscript.

## Competing interests

The authors declare no competing interests.
