## [Peer Review file · Nature Communications]

Gastruloids are competent to specify both cardiac and skeletal muscle lineages.

Corresponding Author: Dr Fabienne Lescroart

Version 0:

Reviewer comments:

Reviewer #1

(Remarks to the Author)

Argiro et al use a combination of staining, lineage tracing and transcriptional analysis to study cardiopharyngeal mesoderm development and specification of cardiomyocyte and skeletal muscle lineages in gastruloids. The work is well done and clearly presented, and results are interesting and novel for the field, especially for what concerns the emergence of skeletal myoblasts, which represent the most innovative aspect of the manuscript. The possibility to study skeletal muscle emergence could indeed open the possibility to guide gastruloid development towards later stages of head or trunk-specific myogenesis, providing a powerful tool for developmental and disease modelling studies.

However, I suggest some points are addressed before publication. In particular, I believe the authors should invest more efforts into the characterization of the emergence of skeletal myoblasts and their origin, considering that this is one of the major strength points of this work.

- In Fig 1b, the beating portion should be highlighted (Day 6 and 11), to provide the reader with a reference.
- The gastruloid morphology at day 11 is less clear than the one at day 6. Did the authors check for the expression of anterior/posterior markers to correctly identify the axis?
- An important claim of the manuscript is the protocol extension until day15. However, little data on gastruloid morphology and cell composition is presented to support this claim. The authors should provide a more in depth characterization of gastruloids beyond day7, including brightfield imaging, whole mount IF and possibly transcriptional analysis of Day15 gastruloids. It would be relevant to show that gastruloids at these late stage in development still maintain the correct polarity and spatial organization which characterize gastruloids, and to understand whether specific cell types are lost or reduced over time (for example, scRNAseq data seem to suggest a reduction in endodermal cells over time)
- N values are missing in figure legends
- It would be very useful if the authors could show whole mount immunofluorescence of the most significant markers, to understand the spatial distribution in gastruloids in 3D
- *Mesp2* has been reported to specifically mark the somitic differentiation front in Day 5 gastruloids (van den Brink et al., 2020). The authors show that at day 4, *Mesp2* is only expressed at very low levels in gastruloids (Fig.S4). How is the expression of *Mesp2* at day5?
- In general, the display of the scRNAseq data could be improved to make the reading easier. For instance, it would be useful to show gene expression in an integrated dataset including gastruloid cells at all time points, to have a better visualization of how marker genes change in the different clusters over time (performing clustering on the integrated dataset)
- Page 8: where is the expression of *Epcam* and *Sox17* in cluster 20 shown? How where the cell identities specifically assigned to visceral endoderm or gut? As the authors highlight in the discussion that endodermal cell perturbations might impact cardiac morphogenesis, it would be useful to include more data on the expression of endodermal markers.
- How do the authors explain the very low levels of *Tbx1* in the scRNAseq datasets in Day6 gastruloids (Fig S8)? This could be better discussed in the manuscript.
- The authors mention that mESCs derived from *Mef2c-Cre*; *Rosa-tdTomato* form beating gastruloids without notable spatial organization. In keeping with this, Fig.3h shows a rather non-polarised structure at day10, where the majority of cells look positive for the shown markers. How are these gastruloids looking at earlier time points? Did the authors try to titrate the Chir concentration during the pulse to see if this would lead to an improved morphology?
- In Fig.4n, the authors should show the co-expression of *Pax3* and *Tbx1* with specific myogenic markers in single cells
- In the reconstruction of transcriptional trajectories, the distance between the time points analysed (Day6 to Day11) might be too big to draw accurate trajectories. This is probably why the fate probability computation towards macrostates described in Figures 5b and c mostly points at trajectories within the same time point.

- In the analysis of branches M and H in SF10, the authors suggest that they might contain juxta-cardiac/proepicardial organ cells based on the expression of Gata6, Hand1, Mab21l2, Tbx18 and Wt1. However, these markers seem not to be specific to these branches, with other branches being equally positive
- The authors suggest that the Pax3 positive myoblast population at day 11 is of somitic origin. As the emergence of skeletal myoblasts is the major novelty point of this paper, the authors should provide more efforts to try to trace back the origin of these cells. When are the lineage decisions (cardiopharyngeal vs somitic) made in gastruloids? A closer time point would probably facilitate this analysis. Do their Mef2c-Cre;Rosa-tdTomato gastruloids generate (Mef2c-Cre;Rosa-tdTomato negative) Pax3+ myoblasts? Did the authors try to promote somitic development by matrigel embedding as described in van den Brink et al., 2020, to see if the number of Pax3+ somitic myoblasts increase?
- In the discussion, the authors hypothesize that heart morphogenesis in their study might be perturbed by the lack of proximity with endodermal structures. This hypothesis could be easily supported with immunostainings and could provide important evidence on the need of this interaction in gastruloids, similarly to embryos.

Minor points:

Page 3 raw 67: compare should be compared

Page 5 raw 133: optimized a previous protocol

Page 9 raws 287-291: this sentence isn't conclusive

Page 13 raw 430: became should be becomes

Page 16 raw 531: I believe the authors refer to Corning 7007 plates

Page 17 raw 537: it would be useful to specify the orbital shaker model and orbit

Page 21 raw 687: wrong superscript

Page 28 Raw 988: recapitulates should be recapitulate

Reviewer #2

(Remarks to the Author)

The manuscript entitled “Cardiopharyngeal mesoderm specification into cardiac and skeletal lineages in gastruloids” uses scRNAseq analysis in combination with whole mount imaging in differentiating gastruloids to provide a thorough description of the emergence of several cardiac and skeletal cell types, which can both be derived from cardiopharyngeal mesoderm during development. The findings show that indeed, cardiopharyngeal mesoderm is present in early gastruloids, and both cardiomyocytes and skeletal myocytes are found subsequently, in later stage gastruloids, thus representing the first report to illustrate and characterize the co-emergence of these two related but distinct lineages. The data collected over several key time points are then integrated for lineage trajectory analysis, which highlight distinct specification trajectories for each lineage, and their relation to early mesoderm populations.

The manuscript is well written and the data are convincing and of interest to both the emerging in vitro model system field as well as the field of developmental biology and early lineage specification. Specific strengths of the study include the excellent introduction to the work, the high quality and rigor of the data provided, the clarity of data presentation (excellent schematics accompanying all of the data in the figures, good description of the data in the legends), the thorough methodology and the thoughtful interpretation and discussion of the findings. The study further stands out for its comprehensiveness, including many comparisons between the in vivo embryos and in vitro gastruloid system at different stages, and the in depth temporal analysis across gastruloid formation. Conceptually, the study provides evidence that cardiac and skeletal lineages can co-develop in the same differentiation protocol and same structure (gastruloids), likely from a shared progenitor population. The scRNAseq data will provide extensive future opportunities to study the mechanisms of cardiac and skeletal lineage specification, particularly in the context of their co-emergence, and will provide a platform to study various perturbations of these process. Some minor comments to consider are listed below.

1. Reproducibility of cell type generation and their frequencies in mouse gastruloids:

-The data illustrating the appearance of skeletal myocytes in mouse gastruloids are convincing, and correlate well between the scRNAseq analysis and the spatio-temporal RNAscope and HCR analyses. Given the relatively low frequency of both the differentiated cardiac and skeletal myocytes (Figure 3i, l and Figure 4o), a quantification across either different organoids, different differentiations or different cell lines would be helpful to understand how robustly the current model generates these cell types (any one of these should be sufficient for the purpose of the present study). In line with the premise of the study, the frequency of co-emergence of both cardiac and skeletal myocytes in gastruloids should be provided.

-Using HCR analysis for both imaging and flow cytometry is an innovative approach and provides important quantitative data. Given the novelty of studying the skeletal component of gastruloids, complementing the RNA analysis with some select protein stains for skeletal myocytes, either by IF or flow cytometry, would further strengthen the overall findings.

-The beautiful temporal profiling at the early stages of the gastruloids suggests that several mesoderm populations are specified robustly and to high frequency (Figure 2). The analysis of day 11 gastruloids suggest that mesoderm persists for a long time, and that the efficiency of specifying mesoderm to the differentiated cell types such as cardiac and skeletal myocytes is comparably low. This may suggest that the current differentiation protocol which is simple and focuses on the early induction, may harbor tremendous potential for additional modifications that could make these later stages more efficient (additional of retinoic acid to specify atrial and nodal CM fates, modifications or screens to enhance production of various skeletal lineages ect). While such strategies are much beyond the scope for the present study, maybe the authors could add a paragraph to the discussion with thoughts and potential strategies for future studies to pursue with this organoid model.

2. Generation of cardiomyocyte subpopulations:

-Consistent with previous studies the authors identify cardiomyocyte subpopulations in the day 11 gastruloids (Figures 2g/h and 3). Both the scRNAseq analysis and validation in gastruloids in comparison with the embryo are well done and quantitative (see comment on reproducibility above). The authors add an excellent cautionary note in their discussion, correctly stating that without any additional analysis, the marker expression in in vitro systems can be challenging to interpret, particularly with respect to assigning subtype specifications. For example, Myl2 is expressed much alter compared to other myosins, and its absence could easily reflect an immature state rather than a non-ventricular fate.

-Figure3: in panel a, red dots are used to identify the 3 different clusters. Red is also used to indicate genes that are most differentially expressed. This data presentation is somewhat misleading and suggestive of the red dots representing the genes next to the UMAP for each cluster. Additionally, it is not specified from which specific comparison these differentially expressed genes were identified.

-In line of the specific CM derivatives of cardiopharyngeal mesoderm it would be interesting to probe the ventricular cardiomyocytes for markers of left/right ventricular CMs.

3. Mechanism of generating 2 skeletal myocyte populations in gastruloids: Skeletal lineage:

Based on expression of Myf5 and the markers of more differentiated skeletal myocytes, cluster 16 seems to contain less differentiated cells compared to cluster 18. This is further supported by the URD trajectory analysis. Interestingly, even though UMAP plots tend to place cells with shared identity closer to one another, the two myogenic clusters are rather far apart, separated interestingly by several mesoderm populations (Figure 4a).

-Proliferation is a common confounder in scRNAseq analysis. With respect to the two myogenic populations, but also all of the scRNAseq analysis presented, have the authors regressed out proliferating cells to test how/if that affects the identity of the different cell populations. Alternatively, rather than removing proliferating cells, which in a developmental context can be essential to keep in the analysis, one could perform cell cycle analysis of the different data sets. One would expect more immature cells to preferentially be in a cycling state, while a sub-set of the differentiated cardiac and skeletal myocytes would be expected to have exited the cell cycle.

-Given the distinct separation of clusters 16 and 18, a DGE analysis might be of interest to understand any additional differences in these 2 cell types.

Minor comment:

-Some of the genes expected to be expressed in the majority of cells within certain clusters (Isl1, Tbx1, others) appear to be expressed only in a relatively small number of cells. Many of these are transcription factors, which tend to be expressed at overall lower levels. Adding the read depth of the data, such as the average number of called genes in the scRNAseq data at the different time points would help with the interpretation of some of these patterns.

Reviewer #3

(Remarks to the Author)

Title: Cardiopharyngeal mesoderm specification into cardiac and skeletal muscle lineages in gastruloids

Authors: Laurent Argiro et al.

Summary: This manuscript uses a gastruloid model to determine whether it is a good system to generate cardiopharyngeal mesoderm (CPM) with cardiac and cardiopharyngeal skeletal muscle derivatives. They generated gastruloids and analyzed them from Day 4 through Day 15. They used a method previously reported by Beccari et al, 2016 and Rossi et al, 2021. They identified cardiomyocyte muscle precursors and skeletal myoblasts after Day 4. This is the first report of the presence of skeletal myoblasts (cardiopharyngeal or somitic) in a gastruloid model without additional chemical treatments. The identification of such skeletal muscle cells is more obvious at later stages, such as Day 11. Much of the analysis is done by single cell RNA-sequencing of Days 4, 5, 6 and 11, in which they first integrate the data with a known atlas of early mouse embryogenesis by Pijuan-Sala et al, 2019. Then they examine molecular markers for both cardiac and skeletal muscle, in which they suggest that the gastruloids can recapitulate the branching between these 2 bipotential cell fates.

Major points:

The gastruloid model has been reported before, but what is new is the length of time the gastruloids are in culture and the demonstration that skeletal muscle progenitor cells can form in gastruloids. The scRNA-seq analysis at four stages focusing on the CPM, rather than it being just an atlas of all cell types.

Overall, this work is an honest indication that the gastruloids are a fair model of CPM specification and differentiation. The authors provide data supporting the strengths and also describe the limitations in the model. They indicate that more work needs to be done using additional lineage tracing and mutants to show that they actually can robustly generate the CPM dual progenitor cell, branchiomic muscle progenitors.

What would overall improve the manuscript is the utilization of similar markers in each comparison along with a sense of the relative expression levels of the CPM marker genes at each stage, or when combined. Also, we don't have a clear sense of how this data compares with previous studies of the Mesp1 lineage identifying the anterior vs posterior second heart field. It is also a bit confusing why, Tbx1, one of the main CPM genes is hardly expressed in the gastruloids.

For consideration, it may be that parallel differentiation for robust CPM skeletal muscle formation would require tweaking of the cell culture medium with growth factors, since in vivo, the environment is somewhat different for the cardiac/skeletal

muscle fate trajectories. Can the medium with growth factors for gastruloids help cause more robust expression/relatively more cells of the CPM skeletal progenitors? Which ones might be particularly useful, esp since there have already been some studies of gastruloids or cardioids in vitro.

What would add a lot would be to understand how loss of particular CPM genes, such as *Isl1* would affect the fate trajectories in the gastruloids (beyond the scope of this manuscript).

Line 133: Please indicate how the gastruloid protocol was optimized in order to culture them for longer times than previously reported. This seems to be critical to distinguish this manuscript from the previous ones.

Figure 1: It is not clear whether CPM and downstream myogenic programs are robustly activated in the gastruloids because the myogenic populations are relatively small. Some clarification of the data would be most helpful:

- Please add the equivalent mouse embryo stages (from Supplementary Figure S2) to Figure 1a.
- Also, some genes are validated in Figure 1g-p, but not all the same genes for all stages, making it confusing to interpret. (with the caveat that some aren't expressed)
- Figure 1d-f, add the expression levels of these genes, besides the fold change, or raw data, even if in a supplemental table of the replicates. For example, it looks like *Tbx1* is hardly expressed in this gastruloid CPM system e.g. Figure 1j. More obvious expression would also help interpret the results of g-p.
- Further, a whole mount analysis of the embryos at E7.75-10.5 would add to the interpretation (if possible, also of the gastruloids). The cell types in the images from the embryos need to be labeled as well.
- According to line 145, *Tcf21* is expressed at Day 4, but in Figure 1f, it looks hardly expressed by qRT-PCR, and its expression isn't shown till Day 7 in Figure 1k-l.
- Looks like *Myf5* and *MyoD* are hardly expressed in the embryo at E9.5 and E10.5. It's hard to see their expression in the equivalent gastruloids.
- Are the beating cells in Figure 1i-p, first heart field versus second heart field cardiomyocyte progenitors? Are any posterior SHF genes expressed, such as *Tbx5*? There is no mention of the pSHF, and showing *Tbx5* would be useful in Figure 1 to identify the first heart field.
- Supplementary Figure 2, or some results from the Pijuan-Sala et al, 2019 paper, should come before the section starting on line 152 (Comparison with mouse embryo shows similar spatio-temporal gene expression), because we need this to know which day of culture of gastruloids correspond to which in vivo mouse embryo day.

Figure 2: Some of the conclusions drawn from comparing a known scRNA-seq atlas from mouse embryos to gastruloids needs clarification.

- Five marker genes for each cluster in Figure 2, from Supplementary Table 1 would make it possible to know how the clusters were named. For example, if *Lhx2*, *Tcf21*, *Tbx1*, *Foxc2*, etc are markers of the early CPM, then we should see their expression in the clusters named as CPM. It may be reasonable to refer to Lescroart et al., Science 2018, where CPM genes were robustly expressed in the *Mesp1* lineage at these same time points. There are also other datasets of the *Mesp1*Cre lineage that can be used.
- Please provide the complete list of marker genes for each cluster from each single cell experiment (in multiple tabs). Is it necessary to generate so many clusters for each stage?
- Violin plots would be useful in Figure 2, as well as similar genes for each stage, for a direct comparison.
- Supplementary Figure 2, a, d, g: its difficult to observe the gastruloid cells within these panels, it would be helpful to show the relative comparisons as a separate panel. It appears, in c, f, i, l, that the gastruloids comprise only a small subset of the cells in the embryo, if this interpretation is correct. It would be useful here to also show the CPM marker genes themselves for consistency. As well as FHF genes such as *Tbx5*.
- There are other atlases that include later staged embryos that they could use to integrate with Day 11 gastruloids (see comment just above).
- Lines 213-215: Please explain more about why *Gata4/6* are expected to be expressed in most clusters, and what their role is in CPM specification. From the references, it seems that they are supposed to restrict CPM specification, at least for the head muscles.
- Lines 223, 231: please provide markers for pharyngeal mesoderm in Figure 2c (see above, first comment for Figure 2).
- Note that *Ebf3* is more broadly expressed than the CPM (Figure 1k).
- Line 240: Neural crest cells were mentioned, but then discussed later that these cells are not present in gastruloids, please comment.
- It would be helpful to provide the specific marker genes used to define cluster types in Supplementary Figures 3, 5 and 7, or indicate which ones are from Supplementary Figures 4, 6, or 8.

Figure 3: Please identify or distinguish if possible the first heart field, versus anterior second heart field versus posterior second heart fields as compared also to the precursors of the proepicardial organ, somatic mesoderm and epicardium, versus endocardium (also see Figure 1).

- Looks like each of the cardiomyocyte clusters are quite heterogenous in composition, is this informative information?
- The *Mef2c*-AHF-Cre lineage study is a good experiment. In the discussion it states that there are some differentiation limitations. However, it does suggest that a significant proportion of cells are derived from this, traditionally anterior second heart field lineage. This can be expanded.

Figure 4: Provide the relative proportions and number of cells in clusters 16 and 18 versus other CPM clusters. It appears as they are very small sized populations relative to the others. Does this imply that the formation of skeletal muscle is not optimal in gastruloids?

- Line 314: *Tcf21* is strongly expressed in the branchiomic muscle progenitors at E10.5, but the statement indicates that it is

not expressed at Day 11, because it is downregulated.

-Is the conclusion correct that both CPM and somitic skeletal muscle cells are mixed together in the same population? This would make the usage of a CPM specific model for branchiomic myogenesis complicated.

Figure 5: In Figure 5a, it looks as though the cells of the individual days of culture of gastruloids do not overlap in terms of cell type, is this the correct interpretation?

-Figures 5b and c do not appear to be particularly useful especially since day 4 should be darkest purple in Figure 5b.

Similar, Figure 5c seems not be particularly helpful.

-Figure 5d, if the cell maps in 5k and n support the author's suggestions. However, it would be good to see some CPM markers on the trees. Such as ones that may be important (or even novel) that specify the CPM lineages, such as in trees D, G vs F.

Figure 6: Not sure what these gastruloids provide besides what is already known about CPM fates. Perhaps there were some novel gene expression dynamics not provide in the figures that can be added? This model figure can be greatly improved. For example, why does their gastruloid system work, when others have not succeeded, is it just cause they waited a few more days?

Minor points:

Figure S1: Add all the main CPM genes *Tbx1*, *Lhx2*, *Pitx2*, *Msc*, *Myf5*, *MyoD*, *Mef2c*, etc, would be helpful.

Not sure why protein names are not in capital letters, e.g. *Mesp1* vs *MESP1*, when referring to the protein. Perhaps this is acceptable as an alternative. Sometimes, the gene is referred to but it's not italicized.

Line 43: define the "22q11.2DS mouse model" is it a deletion or gene mutation?

Line 49: It may be more efficient to use CPM for "cardiopharyngeal mesoderm" once its defined to save space.

Lines 51 and 67: would be helpful for those in the field, to refer to the CPM muscle progenitors as branchiomic muscle progenitors. What about extraocular muscles, are there markers for those in the data?

Line 69: include *Pax7* somewhere, since its used later to mark somitic muscle precursors.

Line 84: *Ciona* should be italicized.

Line 95: Cardiopharyngeal doesn't need to have "C" in a capital letter. It's in the middle of the section title.

Version 1:

Reviewer comments:

Reviewer #1

(Remarks to the Author)

The manuscript by Argiro and colleagues has significantly improved following the recent revisions, with enhanced quality and depth of data. The claims are now supported by clearer experimental evidence and quantifications. Most revision points have been addressed, making the manuscript in my opinion suitable for acceptance. However, there is one concern that I still find in the current version of the manuscript, and which was not fully addressed by the revision. The authors assert an optimization of the protocol (lines 121-126) compared to previous publications, yet no significant new steps are specified. This could mislead readers, so I recommend revising this sentence to accurately reflect the changes. Additionally, while gastruloids survive longer, crucial gastruloid features seem to diminish over time, as evidenced by the loss of posterior tissues and a reduction in the elongation index (though not specifically measured). It appears the authors have extended the growth timeline, sometimes at the expense of axial elongation, favoring anteriorization. This allows for the capture of later-stage differentiation events. I thus suggest that the optimization claim is removed, and emphasis is placed on the need for a longer timeline for differentiation, even if it compromises gastruloid morphology.

As a minor note, for easier comparison, it would be beneficial if the scale of the scRNAseq time course data plots were consistent across time points (Fig. 3, Supplementary Fig. 5).

Overall, this paper will be a valuable resource for future studies, and I look forward to future developments in lineage tracing and further differentiation, as mentioned in the authors' response.

Reviewer #2

(Remarks to the Author)

In their revised manuscripts, Argiro and colleagues provide a substantial amount of new and relevant data on their already excellent original study of co-developing cardiac and skeletal myocytes in a stem cell-derived gastruloid model. The new information provided includes new data analysis, new quantifications of different cell types in the gastruloids, and new marker analysis. These data collectively provide many helpful clarifications and resulted in an improved overall manuscript. The

authors have also added additional details on some of the methodology, and they have carefully revised the writing in all sections according with the new data. Lastly, the authors have expanded their discussion to include information that will be helpful to readers with and without previous expertise on this model system. All of my comments were addressed as listed specifically below.

1. Reproducibility of cell type generation and their frequencies in mouse gastruloids:

-The revised manuscript now includes a number of quantifications of cardiac and skeletal cells in different organoids from multiple differentiations. Some of the quantifications remain at a rather high level (counting beating organoids), which is a good start and can be expanded to quantification at the cellular level (percentage of each cell type per organoid and differentiation) in future studies.

-The MyoG antibody stain for skeletal muscle cells, which was added as an entirely new assay, nicely supports the HCR analysis of the original submission. In addition to strengthening the study, this will collectively provide a comprehensive set of tools for adopters of this organoid model moving forward.

-The authors added a helpful and informative paragraph with their thoughts on potential strategies to further develop the new protocol.

2. Generation of cardiomyocyte subpopulations:

The revised manuscript contains a large amount of new phenotyping data for several cardiomyocyte subtypes, via new HCR analysis. These data suggest that indeed, there seem to be multiple cardiomyocyte sub-types present in the organoids, which closely resemble their in vivo counterparts in the mid-gestation embryo heart. The analysis of left-right identity is particularly interesting, and the lineage-tracing tools incorporated in the organoid model are innovative and state-of-the art to look at these questions.

3. Mechanism of generating 2 skeletal myocyte populations in gastruloids:

The newly performed cell cycle analysis has revealed interesting information on the seemingly unexpected separation of the two skeletal myocyte populations, indeed suggesting that cell cycle status is distinct in each. I find this interesting as it suggests that at this stage of the organoid protocol, organoids may contain different developmental stages of the skeletal myocyte lineage, and having these data available to query this further will might reveal mechanisms involved in skeletal myocyte differentiation.

Minor comments:

All addressed

Reviewer #3

(Remarks to the Author)

This is a beautiful paper that is significantly improved since the original version. In this paper they show that gastruloids that are cultured to day 11, can have both skeletal muscle and cardiac cell types, which are the 2 different trajectories of the cardiopharyngeal mesoderm. They have greatly improved the manuscript by including quantification, adding expression data from embryos, to compare to gastruloids and better explains the strengths and limitations of analyzing gene expression in gastruloids versus single cell RNA-seq.

Minor comments:

1) The 22q11.2DS model mentioned on line 42 is a Tbx1 mutant, not a deletion mouse model, in that reference.

2) Tbx1 is a key gene in the CPM, and they show it is nicely expressed in Figure 2n, despite very little expression in the single cell data. They might want to indicate the percent cells expressing Tbx1 by qRT-PCR or expression analysis.

3) In Figure 5n, the + sign is hard to see for Myf5 (f) and Myod (g) expression. Maybe the images can be enlarged.

Point-by-point response to the reviewers' comments

Reviewer #1 (Remarks to the Author):

Argiro et al use a combination of staining, lineage tracing and transcriptional analysis to study cardiopharyngeal mesoderm development and specification of cardiomyocyte and skeletal muscle lineages in gastruloids. The work is well done and clearly presented, and results are interesting and novel for the field, especially for what concerns the emergence of skeletal myoblasts, which represent the most innovative aspect of the manuscript. The possibility to study skeletal muscle emergence could indeed open the possibility to guide gastruloid development towards later stages of head or trunk-specific myogenesis, providing a powerful tool for developmental and disease modelling studies. However, I suggest some points are addressed before publication. In particular, I believe the authors should invest more efforts into the characterization of the emergence of skeletal myoblasts and their origin, considering that this is one of the major strength points of this work.

We thank the reviewer for this positive assessment of the manuscript. As suggested, we have worked on the characterization of the late staged gastruloids and on the origin of skeletal myogenesis to improve our manuscript.

- In Fib 1b, the beating portion should be highlighted (Day 6 and 11), to provide the reader with a reference.

We thank the reviewer for this insightful comment. We now have highlighted the beating areas at day 7 and 11 (Fig. 1b) and included a video of beating gastruloids (Supplementary Video 1).

- The gastruloid morphology at day 11 is less clear than the one at day 6. Did the authors check for the expression of anterior/posterior markers to correctly identify the axis?

The reviewer raised an important point. This is true that we had not characterized in detail these late gastruloids. We have now included RNAscope experiments at day 7 and day 11 with *Hoxc4* probe as a posterior marker. This is now included in Supplementary Fig. 1e-f. Interestingly we noticed an anti-correlation between the expression of *Hoxc4* and *Tnnt2*, such that *Hoxc4* expression domain is extended when *Tnnt2* domain is reduced and, on the opposite, *Hoxc4* expression is reduced when gastruloids showed high *Tnnt2* expression. This point is now discussed in the text (lines 133-140).

- An important claim of the manuscript is the protocol extension until day15. However, little data on gastruloid morphology and cell composition is presented to support this claim. The authors should provide a more in depth characterization of gastruloids beyond day7, including brightfield imaging, whole mount IF and possibly transcriptional analysis of Day15 gastruloids. It would be relevant to show that gastruloids at these late stage in development still maintain the correct polarity and spatial organization which characterize

gastruloids, and to understand whether specific cell types are lost or reduced over time (for example, scRNAseq data seem to suggest a reduction in endodermal cells over time)

In the revised manuscript, we have further characterized in more detail gastruloids after day 7. In addition to the antero-posterior axis, we now also include characterization of the composition of the gastruloids with immunofluorescence. We have investigated the expression of cTnT (cardiomyocytes), VEGFR2 (endothelial cells) and E-cadherin (endoderm) at day 7 and day 11. This is now provided in Supplementary Figure 1a and c and discussed in the manuscript in the first paragraph of the results section (lines 129-133). We have not observed a major reduction of the endoderm as the scRNAseq experiments suggested it.

We showed that there were no major differences between day 11 and day 15 (as shown below for cTnT, VEGFR2 and E-cadherin, or for *Myh3* and *MyI7*). We decided to only present data on day 11 in the manuscript since the N values were less important on day 15 and not so relevant.

• N values are missing in figure legends

Sorry for the omission. This has now been added in each figure legends.

• It would be very useful if the authors could show whole mount immunofluorescence of the most significant markers, to understand the spatial distribution in gastruloids in 3D

We have performed several new wholemount RNAscope, immunofluorescence and HCR along the manuscript including experiments with cardiopharyngeal mesodermal markers (Fig. 2). We have also included better characterization of the cardiomyocytes cell types and skeletal myoblasts with wholemount immunofluorescence, RNAscope or HCR (Fig. 4 and 5).

- Mesp2 has been reported to specifically mark the somitic differentiation front in Day 5 gastruloids (van den Brink et al., 2020). The authors show that at day 4, Mesp2 is only expressed at very low levels in gastruloids (Fig.S4). How is the expression of Mesp2 at day5?

We have now included Mesp2 expression at day 5 (Supplementary Fig. 9). Mesp2 is also expressed at low levels in gastruloids at day 5.

- In general, the display of the scRNAseq data could be improved to make the reading easier. For instance, it would be useful to show gene expression in an integrated dataset including gastruloid cells at all time points, to have a better visualization of how marker genes change in the different clusters over time (performing clustering on the integrated dataset)

We indeed have considered the point raised by the reviewer. In fact, we decided on purpose to analyze the datasets corresponding to the different time points separately and to only merge the different datasets in the analyses corresponding to Fig. 6a without performing a statistical integration. Indeed, the different time points correspond to different sequencing batches. In this context, a statistical integration (e.g., using the Seurat IntegrateData function...) would correct the batches, but also the true biological signal corresponding to the differences between time points. We hence decided to apply the analysis pipeline, including the clustering step, independently to each dataset.

To allow better visualization of how marker genes change over time, we have modified the original Figure 2 (now Fig. 3) and added a new supplementary figure (Supplementary Figure 5) to visualize gene expression with UMAP or violin plots with the same key markers at different time points.

- Page 8: where is the expression of Epcam and Sox17 in cluster 20 shown? How where

the cell identities specifically assigned to visceral endoderm or gut? As the authors highlight in the discussion that endodermal cell perturbations might impact cardiac morphogenesis, it would be useful to include more data on the expression of endodermal markers.

We performed a cell type annotation transfer as described in the Seurat tutorial “Mapping and annotating query datasets”. We used the atlas of Pijuan-Sala et al. as reference. In concrete terms, the data are integrated with the atlas as shown in Supplementary Fig. 4. Every data cell is labelled with the same cell type as the nearest cell (in terms of expression profile) from the atlas in terms of its expression profile.

To clarify this point in the manuscript, we have added the UMAP for *Epcam* expression on day 11 in Supplementary Fig. 12.

• How do the authors explain the very low levels of *Tbx1* in the scRNAseq datasets in Day6 gastruloids (Fig S8)? This could be better discussed in the manuscript.

We agree with the reviewer’s comment that *Tbx1* expression is very low in our scRNAseq datasets. We first improved the representation of the scRNAseq data to highlight better positive cells (putting positive cells on top layer of the UMAP). On the other hand, we believe that the low sequencing saturation (about 20% for day 4 and day 5 and about

40% for day 6 and day 11) might be why transcription factors expressed at low levels such as *Tbx1* might show low expression in scRNAseq dataset.

However, both qPCR and RNAscope experiments show robust expression in gastruloids (Fig. 2d, 2g-h and 2n), we are thus confident that *Tbx1* is expressed in gastruloids. We have included the sequencing saturation in the Materials and Methods section and included a sentence of the level of *Tbx1* in the scRNAseq in the discussion (lines 450-452): “Expression of transcription factors, such as *Tbx1*, is however relatively low in our scRNAseq analysis. This could be explained by the low sequencing saturation (see Methods), since RNAscope experiments have confirmed the expression of such key cardiopharyngeal markers in gastruloids.”

- The authors mention that mESCs derived from Mef2c-Cre; Rosa-tdTomato form beating gastruloids without notable spatial organization. In keeping with this, Fig.3h shows a rather non-polarised structure at day10, where the majority of cells look positive for the shown markers. How are these gastruloids looking at earlier time points? Did the authors try to titrate the Chir concentration during the pulse to see if this would lead to an improved morphology?

We want to thank the reviewer for the insightful comment and suggestion. We have observed that this particular mESC line (derived directly from mouse blastocysts) was really sensitive in general, and to Chiron concentration in particular. We have performed several experiments and observed few gastruloids with non-polarised structures (with 3 μ M Chiron concentration) as previously shown in the original manuscript, while other gastruloids treated with 1 μ M Chiron concentration have better polarization (see below and Fig. 4n-o). We have added a sentence in that direction in the Materials and Methods section (lines 606-609).

Here is what happen earlier than day 7 (with a pulse of 3 μ M Chiron).

- In Fig.4n, the authors should show the co-expression of Pax3 and Tbx1 with specific myogenic markers in single cells.

We thank the reviewer for this relevant comment. We have added the expression of *Myf5* and *Myod* to that panel which clearly make the point about myogenesis from *Pax3* and *Tbx1*+ progenitors. This is now added in panel (Fig. 5f-g). In addition, we have now included a representative image of wholemount RNAscope with *Tbx1*, *Pax3* and *Myf5* probes, showing independently that gastruloids at day 11 contains *Tbx1*+*Myf5*+ and *Pax3*+*Myf5*+ muscle precursors (Fig. 5k).

- In the reconstruction of transcriptional trajectories, the distance between the time points analysed (Day6 to Day11) might be too big to draw accurate trajectories. This is probably why the fate probability computation towards macrostates described in Figures 5b and c mostly points at trajectories within the same time point.

We completely agree with this comment. Ideally, it would have been great to collect at every single day up to day 11. Unfortunately, for financial constrains we decided to focus on day 6 and day 11. We added a comment in the discussion to make this point clear in the manuscript (lines 454-457).

The identification of a macrostate relies on the observation of metastability within cellular dynamics, encompassing initial, terminal, and intermediate states. An important factor as well is the proliferative status of the cells. The proportion of cells in the G1 phase begins at 0.25 on day 4, then rises to 0.38 on day 5, and further progresses to 0.45 on day 6, ultimately reaching 0.64 by day 11. The lower proportion of cycling cells at the final time point suggests the presence of metastability in the day 11 data but not in the others. We have performed a study focused on macrostates observed at the terminal time point (Fig. 6a-c), particularly within clusters 8 and 18. By computing fate probabilities associated with these macrostates, we can track cellular lineages. Notably, the probability of cells transitioning into the cardiomyocyte macrostate escalates from cluster 6 through cluster 9, ultimately converging in cluster 8. Similarly, skeletal myoblasts traverse cluster 16 before reaching cluster 18.

- In the analysis of branches M and H in SF10, the authors suggest that they might contain juxta-cardiac/proepicardial organ cells based on the expression of *Gata6*, *Hand1*,

Mab21l2, Tbx18 and Wt1. However, these markers seem not to be specific to these branches, with other branches being equally positive

There are no perfect markers for the juxta-cardiac/proepicardial organ cells and it is true that *Gata6*, *Hand1*, *Mab21l2*, *Tbx18* and *Wt1* expression patterns, in the embryo, are not limited to the juxta-cardiac/proepicardial organ. In the embryo, the proepicardial-organ and its derivatives are identified with the association of these markers and the localization of their expression. Since there is no real morphogenesis in gastruloids, it is difficult to address formally the nature of these branches. We have thus added more nuance in our text (lines 403-404).

- The authors suggest that the Pax3 positive myoblast population at day 11 is of somitic origin. As the emergence of skeletal myoblasts is the major novelty point of this paper, the authors should provide more efforts to try to trace back the origin of these cells. When are the lineage decisions (cardiopharyngeal vs somitic) made in gastruloids? A closer time point would probably facilitate this analysis. Do their Mef2c-Cre;Rosa-tdTomato gastruloids generate (Mef2c-Cre;Rosa-tdTomato negative) Pax3+ myoblasts? Did the authors try to promote somitic development by matrigel embedding as described in van den Brink et al., 2020, to see if the number of Pax3+ somitic myoblasts increase?

We thank the reviewer for the relevant comment. It will indeed be fundamental to address when lineage segregation occurs in gastruloids. This is a question that is not completely addressed in mouse embryos for the cardiopharyngeal mesoderm neither. We will address this point soon, in the near future, in embryos and gastruloids with more appropriate tools (lineage tracing).

As mentioned in the discussion, *Mef2c-Cre;Rosa-tdTomato* mESC line was unfortunately unable to develop skeletal myoblasts, limiting the lineage tracing experiments suggested by the reviewer. However, we have added several additional wholemount multiplex fluorescent *in situ* hybridization, at day 11, which confirmed scRNAseq data, i.e., that muscle precursors arise from both cardiopharyngeal (*Tbx1+/Myf5+*) and somitic (*Pax3+/Myf5+*) muscle precursors. These data are now included in Fig. 5j-k. We have not investigated all time points but noticed that there was still no *Myf5* expression at day 8 and *Myf5* expression is still relatively low at day 10 (data not shown).

We have not tried to promote somitic mesoderm by matrigel embedding, since our experience with matrigel is limited. However, we observed that gastruloids with less beating cells (more posterior) have a broader expression of *Pax3* than those having strong beating. Skeletal myogenesis was found also in these gastruloids (see below). Since the number of gastruloids with such profile was low, we have not included these results in our manuscript. But these results clearly indicate that making gastruloids more posterior fate (with higher dose of Chiron for example) will likely increase the number of *Pax3+* somitic myoblasts.

• In the discussion, the authors hypothesize that heart morphogenesis in their study might be perturbed by the lack of proximity with endodermal structures. This hypothesis could be easily supported with immunostainings and could provide important evidence on the need of this interaction in gastruloids, similarly to embryos.

We have added in Supplementary Fig. 1, new wholemout immunofluorescence with E-cadherin antibody. While there is no quantification, it looks like the endoderm is reduced in our study compared to what has been previously published by Rossi et al. Comparison at day 7 is shown below. We have added a note on this in the discussion (lines 517-519).

Minor points:

Page 3 raw 67: compare should be compared.

We apologize for the mistake. This has been corrected.

Page 5 raw 133: optimized a previous protocol

This has been corrected.

Page 9 raws 287-291: this sentence isn't conclusive

This has been corrected.

Page 13 raw 430: became should be becomes

This has been corrected.

Page 16 raw 531: I believe the authors refer to Corning 7007 plates

This has been corrected.

Page 17raw 537 : it would be useful to specify the orbital shaker model and orbit

This has been added.

Page 21 raw 687: wrong superscript

This has been corrected.

Page 28 Raw 988: recapitulates should be recapitulate

This has been corrected.

-
Reviewer #2 (Remarks to the Author):

The manuscript entitled “Cardiopharyngeal mesoderm specification into cardiac and skeletal lineages in gastruloids” uses scRNAseq analysis in combination with whole mount imaging in differentiating gastruloids to provide a thorough description of the emergence of several cardiac and skeletal cell types, which can both be derived from cardiopharyngeal mesoderm during development. The findings show that indeed, cardiopharyngeal mesoderm is present in early gastruloids, and both cardiomyocytes and skeletal myocytes are found subsequently, in later stage gastruloids, thus representing the first report to illustrate and characterize the co-emergence of these two related but distinct lineages. The data collected over several key time points are then integrated for lineage trajectory analysis, which highlight distinct specification trajectories for each lineage, and their relation to early mesoderm populations.

The manuscript is well written and the data are convincing and of interest to both the emerging in vitro model system field as well as the field of developmental biology and early lineage specification. Specific strengths of the study include the excellent introduction to the work, the high quality and rigor of the data provided, the clarity of data presentation (excellent schematics accompanying all of the data in the figures, good description of the data in the legends), the thorough methodology and the thoughtful interpretation and discussion of the findings. The study further stands out for its comprehensiveness, including many comparisons between the in vivo embryos and in vitro gastruloid system at different stages, and the in depth temporal analysis across gastruloid formation. Conceptually, the study provides evidence that cardiac and skeletal lineages can co-develop in the same differentiation protocol and same structure (gastruloids), likely from a shared progenitor population. The scRNAseq data will provide extensive future opportunities to study the mechanisms of cardiac and skeletal lineage specification, particularly in the context of their co-emergence, and will provide a platform to study various perturbations of these process. Some minor comments to consider are listed below.

We thank the reviewer for this positive assessment. We have addressed most of the comments, which have improved the quality of the manuscript and strengthen our conclusions.

1. Reproducibility of cell type generation and their frequencies in mouse gastruloids:
-The data illustrating the appearance of skeletal myocytes in mouse gastruloids are convincing, and correlate well between the scRNAseq analysis and the spatio-temporal RNAscope and HCR analyses. Given the relatively low frequency of both the differentiated cardiac and skeletal myocytes (Figure 3i, l and Figure 4o), a quantification across either different organoids, different differentiations or different cell lines would be helpful to understand how robustly the current model generates these cell types (any one of these should be sufficient for the purpose of the present study). In line with the premise

of the study, the frequency of co-emergence of both cardiac and skeletal myocytes in gastruloids should be provided.

We thank the reviewer for the relevant and fair comment. We have now included several quantifications to illustrate the reproducibility and robustness of the specification and differentiation of cardiomyocytes and skeletal myoblasts. We have added the percentage of beating gastruloids (Fig. 1c – n=8 independent experiments). Additionally, we have now provided quantification of gastruloids with cTNT, *Tnnt2*, *Myl7*, *Myf5* or *Myh3* expression (Supplementary Figure 1 and Fig. 2o, r, u – n>14 gastruloids in >3 independent experiments).

-Using HCR analysis for both imaging and flow cytometry is an innovative approach and provides important quantitative data. Given the novelty of studying the skeletal component of gastruloids, complementing the RNA analysis with some select protein stains for skeletal myocytes, either by IF or flow cytometry, would further strengthen the overall findings.

We agree that most of our previous results only relied on mRNA expression and not on proteins. We have now included wholemount immunofluorescent of gastruloids with a MyoG antibody (Fig. 5i), confirming the expression of skeletal muscle markers also at the protein level.

-The beautiful temporal profiling at the early stages of the gastruloids suggests that several mesoderm populations are specified robustly and to high frequency (Figure 2). The analysis of day 11 gastruloids suggest that mesoderm persists for a long time, and that the efficiency of specifying mesoderm to the differentiated cell types such as cardiac and skeletal myocytes is comparably low. This may suggest that the current differentiation protocol which is simple and focuses on the early induction, may harbor tremendous potential for additional modifications that could make these later stages more efficient (additional of retinoic acid to specify atrial and nodal CM fates, modifications or screens to enhance production of various skeletal lineages ect). While such strategies are much beyond the scope for the present study, maybe the authors could add a paragraph to the discussion with thoughts and potential strategies for future studies to pursue with this organoid model.

We thank the reviewer for this comment. As suggested, we have added some points to the discussion along this line including potential strategies that could enhance the different cardiac or skeletal muscle fates (lines 469-475 and 496-501).

2. Generation of cardiomyocyte subpopulations:

-Consistent with previous studies the authors identify cardiomyocyte subpopulations in the day 11 gastruloids (Figures 2g/h and 3). Both the scRNAseq analysis and validation in gastruloids in comparison with the embryo are well done and quantitative (see comment on reproducibility above). The authors add an excellent cautionary note in their discussion, correctly stating that without any additional analysis, the marker expression in in vitro systems can be challenging to interpret, particularly with respect to assigning

subtype specifications. For example, *MyI2* is expressed much alter compared to other myosins, and its absence could easily reflect an immature state rather than a non-ventricular fate.

We agree with the caution with marker analysis such as *MyI2*. We have now provided an HCR in a mouse heart at E10.5 to show that at this stage *MyI2* is clearly restricted to ventricular cardiomyocytes (Fig. 4g). We have also strengthened our analysis of the different cardiomyocytes subtype with wholemount HCR of *MyI2* and *MyI7* in *Zx1* and *Mef2c-AHF-Cre/tdTomato* gastruloids (Fig. 4h and n-o). Our results suggest that gastruloids can form cardiomyocytes from the right and left ventricles in addition to atrial cardiomyocytes, and likely also outflow tract cardiomyocytes.

“To validate the presence of atrial and ventricular cardiomyocytes in gastruloids we performed in situ hybridization using the HCR method with specific ventricular and atrial probes. We first showed that at E10.5, MyI2 is indeed restricted to ventricular cardiomyocytes, while MyI7 is expressed in all cardiomyocytes but enriched in atrial cardiomyocytes (Fig. 4g). We then performed HCR experiments in wholemount gastruloids or followed by flow cytometry (see Methods). We found clearly distinct domains of MyI2 and MyI7 expression, each exhibiting a unique but seemingly random organization (n=13/15 - Fig. 4h) or in two well organized domains (n=2/15 - Fig. 4h’).” (lines 285-291).

“We additionally performed in situ hybridization, with the ventricular specific MyI2 probe, to discriminate left ventricular versus right ventricular fates in Mef2c-AHF-enhancer-Cre; Rosa^{tdTomato/+} gastruloids collected at day 11 (Fig. 4n). We observed the presence of both MyI2+/tdTomato+ and MyI2+/tdTomato- domains (Fig.4o), supporting the specification of right and left ventricular cardiomyocytes respectively. MyI2-/MyI7+/tdTomato+ cells also present in gastruloids might correspond to outflow tract-like cardiomyocytes.” (lines 316-325).

-Figure3: in panel a, red dots are used to identify the 3 different clusters. Red is also used to indicate genes that are most differentially expressed. This data presentation is somewhat misleading and suggestive of the red dots representing the genes next to the UMAP for each cluster. Additionally, it is not specified from which specific comparison these differentially expressed genes were identified.

We have modified this panel and the color code for clarity.

-In line of the specific CM derivatives of cardiopharyngeal mesoderm it would be interesting to probe the ventricular cardiomyocytes for markers of left/right ventricular CMs.

As discussed above, we have made use of the *Mef2c-AHF-Cre/tdTomato* mESC line and used *MyI2* probe to distinguish right and left ventricular cardiomyocytes. *Mef2c-AHF* lineage tracing label only the right ventricle (together with the outflow tract). We have observed *MyI2+/tdTomato+* as well as *MyI2+/tdTomato-* cells in *Mef2c-AHF-Cre/tdTomato* gastruloids (Fig. 4n-o), suggesting that gastruloids can form cardiomyocytes with both right and left ventricular identity.

3. Mechanism of generating 2 skeletal myocyte populations in gastruloids: Skeletal lineage:

Based on expression of *Myf5* and the markers of more differentiated skeletal myocytes, cluster 16 seems to contain less differentiated cells compared to cluster 18. This is further supported by the URD trajectory analysis. Interestingly, even though UMAP plots tend to place cells with shared identity closer to one another, the two myogenic clusters are rather far apart, separated interestingly by several mesoderm populations (Figure 4a).

Clustering and UMAP visualization are two separated analyses. Clustering is performed on the data transformed by a linear dimensionality reduction (PCA) and intend to highlight groups of cells with similar gene expression. UMAP is a non-linear visualization and is known to not always preserve distances between cells. Overall, 2D distances in the UMAP plot cannot be directly interpreted as biological distances in the gene expression, contrarily to the clustering.

-Proliferation is a common confounder in scRNAseq analysis. With respect to the two myogenic populations, but also all of the scRNAseq analysis presented, have the authors regressed out proliferating cells to test how/if that affects the identity of the different cell populations. Alternatively, rather than removing proliferating cells, which in a developmental context can be essential to keep in the analysis, one could perform cell cycle analysis of the different data sets. One would expect more immature cells to preferentially be in a cycling state, while a sub-set of the differentiated cardiac and skeletal myocytes would be expected to have exited the cell cycle.

We thank the reviewer for this very relevant comment. In our analysis, we decided not to regress the cell cycle nor remove proliferating cells. The analysis of the cell cycle signal shows that the proportion of cells in the G1 phase begins at 0.25 on day 4, then rises to 0.38 on day 5, and further progresses to 0.45 on day 6, ultimately reaching 0.64 by day 11. Regarding the skeletal myocytes at day 11, we observe that the cluster 16 has 88% of its cells in proliferative state, while 96% cells of the cluster 18 are in G1 phase. Due to the high proportion of the cells in that phase, we can hypothesize that they exited the cell cycle. This is a very relevant information further supporting the idea that cluster 16 corresponds to cells in a precursor state, while cells of cluster 18 are less proliferative and more committed. These novel results, prompted by the reviewer's comment, has now been added in Fig. 5b.

-Given the distinct separation of clusters 16 and 18, a DGE analysis might be of interest to understand any additional differences in these 2 cell types.

A DGE analysis between clusters 16 and 18 was already included (Supplementary Table 3). We have however extended this table.

Minor comment:

-Some of the genes expected to be expressed in the majority of cells within certain clusters (*Isl1*, *Tbx1*, others) appear to be expressed only in a relatively small number of cells. Many of these are transcription factors, which tend to be expressed at overall lower levels. Adding the read depth of the data, such as the average number of called genes in the scRNAseq data at the different time points would help with the interpretation of some of these patterns.

We thank the reviewer for this comment. This is now included in the Methods section and we have also added a comment on the low level of *Tbx1* in the discussion (lines 450-452): *“Expression of transcription factors, such as Tbx1, is however relatively low in our scRNA-seq analysis. This could be explained by the low sequencing saturation (see Methods), since RNAscope experiments have confirmed the expression of such key cardiopharyngeal markers in gastruloids.”*

-
Reviewer #3 (Remarks to the Author):

Title: Cardiopharyngeal mesoderm specification into cardiac and skeletal muscle lineages in gastruloids

Authors: Laurent Argiro et al.

Summary: This manuscript uses a gastruloid model to determine whether it is a good system to generate cardiopharyngeal mesoderm (CPM) with cardiac and cardiopharyngeal skeletal muscle derivatives. They generated gastruloids and analyzed them from Day 4 through Day 15. They used a method previously reported by Beccari et al, 2016 and Rossi et al, 2021. They identified cardiomyocyte muscle precursors and skeletal myoblasts after Day 4. This is the first report of the presence of skeletal myoblasts (cardiopharyngeal or somitic) in a gastruloid model without additional chemical treatments. The identification of such skeletal muscle cells is more obvious at later stages, such as Day 11. Much of the analysis is done by single cell RNA-sequencing of Days 4, 5, 6 and 11, in which they first integrate the data with a known atlas of early mouse embryogenesis by Pijuan-Sala et al, 2019. Then they examine molecular markers for both cardiac and skeletal muscle, in which they suggest that the gastruloids can recapitulate the branching between these 2 bipotential cell fates.

Major points:

The gastruloid model has been reported before, but what is new is the length of time the gastruloids are in culture and the demonstration that skeletal muscle progenitor cells can form in gastruloids. The scRNA-seq analysis at four stages focusing on the CPM, rather than it being just an atlas of all cell types.

Overall, this work is an honest indication that the gastruloids are a fair model of CPM specification and differentiation. The authors provide data supporting the strengths and also describe the limitations in the model. They indicate that more work needs to be done using additional lineage tracing and mutants to show that they actually can robustly generate the CPM dual progenitor cell, branchiomic muscle progenitors.

We thank the reviewer for this positive assessment of the manuscript.

What would overall improve the manuscript is the utilization of similar markers in each comparison along with a sense of the relative expression levels of the CPM marker genes at each stage, or when combined.

We thank the reviewer for this comment. We have performed several new wholemount RNAscope, immunofluorescence and HCR along the manuscript, with close comparison with the embryo. For cardiopharyngeal mesoderm markers, we have included

wholemout RNAscope or HCR (Fig. 2). While it is difficult to test exactly the same markers at all the stages since some are no longer expressed, we have consistently marked *Tbx1* in all stages analyzed.

Also, we don't have a clear sense of how this data compares with previous studies of the *Mesp1* lineage identifying the anterior vs posterior second heart field.

We agree that we had not detailed *Mesp1* lineages in the original manuscript. In its revised version, we have now added few wholemount RNAscope experiments at day 4 and day 5, (with comparison with mouse embryos), describing *Foxc2* and *Hoxb1* expression as markers of the anterior and posterior SHF. This is now added in Supplementary Figure 13.

It is also a bit confusing why, *Tbx1*, one of the main CPM genes is hardly expressed in the gastruloids.

We agree that our initial section on Fig. 1 did not show a very strong expression of *Tbx1*. The revised manuscript now shows different panels with wholemount RNAscope with *Tbx1* probe. This now included in Fig. 2 and Fig. 5.

For consideration, it may be that parallel differentiation for robust CPM skeletal muscle formation would require tweaking of the cell culture medium with growth factors, since in vivo, the environment is somewhat different for the cardiac/skeletal muscle fate trajectories. Can the medium with growth factors for gastruloids help cause more robust expression/relatively more cells of the CPM skeletal progenitors? Which ones might be particular useful, esp since there have already been some studies of gastruloids or cardioids in vitro.

We thank the reviewer for the comment. We have added few points in the discussion on the different strategies to enhance the different cardiac or skeletal muscle fates or to promote cardiopharyngeal vs somitic mesoderm (lines 469-475 and lines 496-501).

“Skeletal myogenesis remains limited in gastruloids, with cluster 16 and 18 comprising only about 4% of all cells (Table 1). Further work is needed to determined how to better promote skeletal myogenesis through the addition of growth factors (HGF, IGF, FGF) or inhibitors (BMP or Rock-inhibitors) (Nandkishore, et al. 2018; Chan et al. 2016). Similarly, in the context of CPM development, it will be important to establish methods to robustly promote CPM at the expense of more posterior/somitic mesoderm. This may be achieved by reducing or inhibiting Wnt and/or Nodal activity, as shown previously in 2D culture (Nandkishore, et al. 2018).”

“It will be also important to play with small molecules inhibitors/activators and/or growth factors to increase or reduce specifically some cardiomyocytes subpopulations. For example, it is conceivable that adding retinoic acid could enhance posterior SHF specification and promote the differentiation of atrial cardiomyocytes.”

What would add a lot would be to understand how loss of particular CPM genes, such as *Isl1* would affect the fate trajectories in the gastruloids (beyond the scope of this manuscript).

We agree with the reviewer that tackling whether and how the loss of cardiopharyngeal transcription factors such as *Tbx1* and *Isl1* affect the fate trajectories is essential. This is our next goal and the gastruloid model represents a good tool to address these questions. We are currently generating several knock-out lines to investigate cardiopharyngeal mesoderm specification in a mutant context.

Line 133: Please indicate how the gastruloid protocol was optimized in order to culture them for longer times than previously reported. This seems to be critical to distinguish this manuscript from the previous ones.

We have now added more details on the extension of the protocol in the result section (lines 125-126). More details are also found in the Methods section.

“After day 7, gastruloids were cultured in N2B27 culture media. Shaking (80-100rpm) was continuous from day 4 to the end of the culture.”

Figure 1: It is not clear whether CPM and downstream myogenic programs are robustly activated in the gastruloids because the myogenic populations are relatively small. Some clarification of the data would be most helpful:

-Please add the equivalent mouse embryo stages (from Supplementary Figure 2) to Figure 1a.

-Also, some genes are validated in Figure 1g-p, but not all the same genes for all stages, making it confusing to interpret. (with the caveat that some aren't expressed)

We have performed several new wholemount RNAscope, immunofluorescence and HCR along the manuscript, with close comparison with the embryo, with additional time points. For cardiopharyngeal mesoderm markers, we have included wholemount RNAscope or HCR (Fig. 2). While it is difficult to test exactly the same markers at all the stages since some are no longer expressed, we have consistently marked *Tbx1* in all stages analyzed.

-Figure 1d-f, add the expression levels of these genes, besides the fold change, or raw data, even if in a supplemental table of the replicates. For example, it looks like *Tbx1* is hardly expressed in this gastruloid CPM system e.g. Figure 1j. More obvious expression would also help interpret the results of g-p.

We have now provided the Source Data with raw data of the qPCR analysis. In the revised manuscript, we have added several wholemount RNAscope with *Tbx1* showing its low but robust expression (Fig. 2 and Fig. 5).

-Further, a whole mount analysis of the embryos at E7.75-10.5 would add to the interpretation (if possible, also of the gastruloids). The cell types in the images from the embryos need to be labeled as well.

Wholemout RNAscope of embryos and gastruloids are now included in Fig. 2, and Supplementary Figure 13.

-According to line 145, *Tcf21* is expressed at Day 4, but in Figure 1f, it looks hardly expressed by qRT-PCR, and its expression isn't shown till Day 7 in Figure 1k-l.
-Looks like *Myf5* and *MyoD* are hardly expressed in the embryo at E9.5 and E10.5. It's hard to see their expression in the equivalent gastruloids.

We agree with the reviewer that *Tcf21* did not seem to be highly expressed at day 4 (qPCR data). This was mostly due to the scale used. The revised figures now show robust expression of *Tcf21* by qPCR (from day 3) (Fig. 1g) and with wholemount RNAscope from day 6 to day 11 (Fig. 2k-l and Fig. 2q). We also agree that *Myf5* and *Myod* expression looked low in our previous figures. We have now included wholemount RNAscope or HCR for *Myf5* and *Myh3* in gastruloids at day 11 and mouse embryos at E9.5-E10.5 with quantifications (Fig. 2p-u). Of note, for technical reason, we have been able to obtain wholemount E9.5 mouse embryos, hybridized with *Myf5* probes (RNAscope) but we were not able to get good hybridization at E10.5. At E9.5, there is not yet expression of *Myf5* in the pharyngeal arches, while skeletal myogenic cells expressing *Myf5* in the somites were detected (Fig. 2p). However, HCR at E10.5 was more successful and we have now included, wholemount in situ hybridization with *Myh3* in the pharyngeal arches of an E10.5 embryo (Fig. 2s).

-Are the beating cells in Figure 1i-p, first heart field versus second heart field cardiomyocyte progenitors? Are any posterior SHF genes expressed, such as *Tbx5*? There is no mention of the pSHF, and showing *Tbx5* would be useful in Figure 1 to identify the first heart field.

We agree with the reviewer that this is a key information. While we have not included *Tbx5* in our revised Figure 2, we have now showed wholemount RNAscope at day 6 and day 11 with *Tbx5* expression (Supplementary Figure 13).

-Supplementary Figure 2, or some results from the Pijuan-Sala et al, 2019 paper, should come before the section starting on line 152 (Comparison with mouse embryo shows similar spatio-temporal gene expression), because we need this to know which day of culture of gastruloids correspond to which in vivo mouse embryo day.

We understand the comment of the reviewer but we have chosen to first show wholemount RNAscope experiments and then the single cell data. We found that this made it smoother to read our manuscript. We have added more time points in Fig2. and we now showed that the comparison with the atlas from the Pijuan-Sala paper match with our results.

Figure 2: Some of the conclusions drawn from comparing a known scRNA-seq atlas from mouse embryos to gastruloids needs clarification.

-Five marker genes for each cluster in Figure 2, from Supplementary Table 1 would make it possible to know how the clusters were named. For example, if *Lhx2*, *Tcf21*, *Tbx1*,

Foxc2, etc are markers of the early CPM, then we should see their expression in the clusters named as CPM. It may be reasonable to refer to Lescroart et al., Science 2018, where CPM genes were robustly expressed in the Mesp1 lineage at these same time points. There are also other datasets of the Mesp1Cre lineage that can be used.

To avoid any bias in the identification of the clusters, the cell type identification was performed by cell type annotation transfer from the Pijuan-Sala et al. atlas (Supplementary Fig. 4) for the data from day 4, day 5 and day 6 (see Methods). This is why those clusters are not named as Lescroart et al., Science 2018. For data of day11, (because the atlas does not cover this stage), we used the TISSUE database with the differentially expressed genes of each cluster.

To allow better visualization of how marker genes change over time, we have modified the original Figure 2 and added a new supplementary figure to visualize gene expression with UMAP (now Fig. 3) or violin plots with the same key markers (*Mesp1*, *Tcf21*, *Myl7*, *Myf5*) at different time points (Supplementary Fig. 5).

-Please provide the complete list of marker genes for each cluster from each single cell experiment (in multiple tabs). Is it necessary to generate so many clusters for each stage?

The top 20 marker genes of each cluster were already included in Supplementary Table 1. We have now extended this table to include a more complete list of markers. Regarding the number of clusters, we used the Leiden algorithm that relies on a single parameter, the resolution. For simplicity and comparison purposes, we decided not to fine-tune this parameter differently for each dataset and rather set the (default) resolution 1 for all the datasets. The number of clusters obtained for each dataset is the one obtained with this parameter setting.

-Violin plots would be useful in Figure 2, as well as similar genes for each stage, for a direct comparison.

We thank the reviewer for the suggestion. To allow better visualization of how marker genes change over time, we have added a new supplementary figure to visualize gene expression with violin plots with the same key markers at different time points (Supplementary Fig. 5).

-Supplementary Figure 2, a, d, g: its difficult to observe the gastruloid cells within these panels, it would be helpful to show the relative comparisons as a separate panel. It appears, in c, f, i, l, that the gastruloids comprise only a small subset of the cells in the embryo, if this interpretation is correct. It would be useful here to also show the CPM marker genes themselves for consistency. As well as FHF genes such as *Tbx5*.

We acknowledge that this Supplementary Figure is difficult to navigate. Our goal was to integrate each single gastruloid time point with the whole embryonic atlas to see where gastruloids cells were located.

To make it clearer:

1. first line: embryonic atlas + day 4
2. second line: embryonic atlas + day 5
3. third line: embryonic atlas + day 6
4. fourth line: embryonic atlas + day 11

1. The first column shows in color only embryonic cells (with colors indicating the stage of collection) and gastruloids cells are shown in grey.
2. The second column shows only embryonic cells (with colors indicating the cell identity) and gastruloids cells are shown in grey.
3. The third column shows only gastruloids cells at the appropriate time point (with colors indicating the cell label based on the embryo atlas). Cells in grey correspond to the embryo.

We have tried to plot gene expression on top of that (see below). It does not really help to make that figure clear.

We have tried to plot gene expression on top of that (see below). It does not really help to make that figure clear.

-There are other atlases that include later staged embryos that they could use to integrate with Day 11 gastruloids (see comment just above).

We agree that there are other atlases with later stages. We kept the same atlas to keep the same reference for comparison of our different time points. Integrating the three first steps with the Pijuan-Sala atlas, and then the last time point with another appeared to be perilous for interpretation.

-Lines 213-215: Please explain more about why Gata4/6 are expected to be expressed in most clusters, and what their role is in CPM specification. From the references, it seems that they are supposed to restrict CPM specification, at least for the head muscles

We thank the reviewer for the relevant comment. We have added some explanations and references on Gata4/6 expression and role in the CPM.

“Gata transcription factors are known to have a broad expression pattern in the anterior/cardiogenic mesoderm as well as in the endoderm (Peterkin et al. 2007). They have a critical function during heart development (Molkentin et al. 1999) and recent work in the fish has shown that they are involved in CPM specification, by promoting cardiac fate and inhibiting pharyngeal identity (Song et al. 2022).”

-Lines 223, 231: please provide markers for pharyngeal mesoderm in Figure 2c (see above, first comment for Figure 2).

To allow better visualization of how cardiopharyngeal mesoderm markers change over time, we have modified the original Figure 2 and added a new supplementary figure to visualize gene expression with UMAP (now Fig. 3) with the same key markers (*Mesp1*, *Tcf21*, *Myf7*, *Myf5*) at different time points. Other markers are found in Supplementary Fig. 7, 9, 11 and 12.

-Note that *Ebf3* is more broadly expressed than the CPM (Figure 1k).

We agree with the reviewer. *Ebf3* is also expressed in the endothelial lineage for example. *Ebf3* has been chosen as it has been shown that Ebf factors were important for CPM specification in tunicates. Among the 4 Ebf in mammals, *Ebf3* is the more specific to the CPM.

-Line 240: Neural crest cells were mentioned, but then discussed later that these cells are not present in gastruloids, please comment.

Neural crest cells are largely absent in gastruloids but we could still detect few *Sox10*+ cells at day 11. We have added a UMAP with *Sox10* expression in Supplementary Fig. 12.

-It would be helpful to provide the specific marker genes used to define cluster types in Supplementary Figures 3, 5 and 7, or indicate which ones are from Supplementary Figures 4, 6, or 8.

Supplementary Fig. 6, 8 and 10 show labels defined by the atlas from Pijuan-Sala et al. We performed a cell type annotation transfer as described in the Seurat tutorial “Mapping and annotating query datasets”. We used the atlas of Pijuan-Sala et al. as reference. Every data cell is labelled with the same cell type as the nearest cell from the atlas in terms of its expression profile. The labels in new Supplementary Fig. 6, 8 and 10 are different from the clustering analysis and are not clusters. Marker genes for these labels are thus defined in the original manuscript (Pijuan-Sala et al).

Figure 3: Please identify or distinguish if possible, the first heart field, versus anterior second heart field versus posterior second heart fields as compared also to the precursors of the proepicardial organ, somatic mesoderm and epicardium, versus endocardium (also see Figure 1).

We have now included in Supplementary Figure 12 additional UMAPs with markers of the SHF (*Isl1*, *Foxc2*). We have also included *Tbx5* which marks FHF-derived cardiomyocytes (when associated with *Tnnt2* expression) or posterior SHF when associated with *Isl1*. Moreover, we have performed several wholemount RNAscope in mouse embryos and gastruloids with some specific markers of the FHF, as well as anterior and posterior SHF (Supplementary Figure 13).

-Looks like each of the cardiomyocyte clusters are quite heterogenous in composition, is this informative information?

The heterogenous composition of the cardiomyocytes clusters is indeed really interesting and informative. Cluster 8 (ventricular cardiomyocytes), for example, likely contains both right and left ventricular cardiomyocytes. We have not performed subclustering of the scRNAseq data and do not discuss this heterogeneity of the clusters but we have extended our analysis and validation of the different subpopulations of cardiomyocytes with wholemount RNAscope or HCR (Fig. 4g-h and n-o).

-The *Mef2c*-AHF-Cre lineage study is a good experiment. In the discussion it states that there are some differentiation limitations. However, it does suggest that a significant proportion of cells are derived from this, traditionally anterior second heart field lineage. This can be expanded.

We want to thank the reviewer for the insightful comment. We have performed additional experiments with flow cytometry and wholemount HCR. We have shown that not all cardiomyocytes are *tdTomato*⁺ showing that as for the embryo, the anterior SHF only contribute to a subpopulation of the all cardiomyocytes (Figure 4). Our additional experiments, with this mESC line, have further suggested that left and right ventricular cardiomyocytes develop in gastruloids.

Figure 4: Provide the relative proportions and number of cells in clusters 16 and 18 versus other CPM clusters. It appears as they are very small sized populations relative to

the others. Does this imply that the formation of skeletal muscle is not optimal in gastruloids?

We agree that this was a missing point. We have added a Table (Table 1) with the number of cells per clusters. We also agree that the proportion of skeletal myoblast is relatively low (about 4%). We have added a point in the discussion on possible strategies to increase the differentiation of skeletal muscle in gastruloids:

“Skeletal myogenesis remains limited in gastruloids, with cluster 16 and 18 comprising only about 4% of all cells (Table 1). Further work is needed to determine how to better promote skeletal myogenesis through the addition of growth factors (HGF, IGF, FGF) or inhibitors (BMP or Rock-inhibitors) (Nandkishore, et al. 2018; Chan et al. 2016). Similarly, in the context of CPM development, it will be important to establish methods to robustly promote CPM at the expense of more posterior/somitic mesoderm. This may be achieved by reducing or inhibiting Wnt and/or Nodal activity, as shown previously in 2D culture (Nandkishore, et al. 2018).”

Clusters	1	2	3	4	5	6	7	8	9	10
Nbr. of cells	1052	934	658	583	538	499	456	444	427	385
Clusters	11	12	13	14	15	16	17	18	19	20
Nbr. of cells	340	296	291	279	257	243	185	67	56	34

-Line 314: Tcf21 is strongly expressed in the branchiomic muscle progenitors at E10.5, but the statement indicates that it is not expressed at Day 11, because it is downregulated.

We agree that our statement was misleading. We have corrected the manuscript. *“In the scRNAseq, Tcf21 was expressed only in the myoblast cluster 16 in a limited number of cells.”* Tcf21 expression has been detected independently in gastruloids with wholemount RNAscope (Fig. 5j).

-Is the conclusion correct that both CPM and somitic skeletal muscle cells are mixed together in the same population? This would make the usage of a CPM specific model for branchiomic myogenesis complicated.

This might have been misleading. CPM and somitic skeletal muscle cells are mixed in the same cluster in scRNA-seq datasets but in the gastruloids, they represent independent cell populations. We have added in Fig. 5 representative images of additional wholemount RNAscope experiments. These results show that Pax3 and Tcf21 or Tbx1 are indeed expressed in different domains of the gastruloid.

Figure 5: In Figure 5a, it looks as though the cells of the individual days of culture of gastruloids do not overlap in terms of cell type, is this the correct interpretation?

It is the correct interpretation as the datasets were merged. Biological and technical information are indivisible. By performing an integration, we were worried to lose biological information.

-Figures 5b and c do not appear to be particularly useful especially since day 4 should be darkest purple in Figure 5b. Similar, Figure 5c seems not be particularly helpful.

In the Figures 6b and 6c, we visualize the fate probabilities of the cells towards the indicated macrostates (cardiomyocytes and skeletal myoblasts, respectively). In contrast with pseudotime, where the color scale represents the timepoints, the gradient represents the probability of the cells to reach the macrostate. For instance, on the Figure 6b, cells from day 4 have a higher probability to reach the macrostate identified in cluster 8 of day11 (Cardiomyocytes) than the cells differentiated into the skeletal myoblasts of day 11 (cluster 18).

-Figure 5d, if the cell maps in 5k and n support the author's suggestions. However, it would be good to see some CPM markers on the trees. Such as ones that may be important (or even novel) that specify the CPM lineages, such as in trees D, G vs F.

We have not found novel markers that specify the CPM lineages and distances between the time points would make this investigation extremely challenging. However, we have plotted the expression of some CPM markers in Supplementary Figure 14.

Figure 6: Not sure what these gastruloids provide besides what is already known about CPM fates. Perhaps there were some novel gene expression dynamics not provide in the figures that can be added? This model figure can be greatly improved. For example, why does their gastruloid system work, when others have not succeeded, is it just cause they waited a few more days?

The manuscript here does not provide novel information on how CPM is specified (but that will be the goal of future work) but rather shows that gastruloids can recapitulates several aspects of CPM and somitic mesoderm development. We have improved our model to show the somitic mesoderm and detail the different cardiomyocytes subtypes found in gastruloids.

Minor points:

Figure S1: Add all the main CPM genes Tbx1, Lhx2, Pitx2, Msc, Myf5, MyoD, Mef2c, etc, would be helpful.

Not sure why protein names are not in capital letters, e.g. Mesp1 vs MESP1, when referring to the protein. Perhaps this is acceptable as an alternative. Sometimes, the gene is referred to but it's not italicized.

We have modified the nomenclature in the text.

Line 43: define the "22q11.2DS mouse model" is it a deletion or gene mutation?

22q11.2 DS is a deletion. We have clarified this point in the manuscript.

Line 49: It may be more efficient to use CPM for “cardiopharyngeal mesoderm” once its defined to save space.

We have now used CPM for “cardiopharyngeal mesoderm” in the manuscript.

Lines 51 and 67: would be helpful for those in the field, to refer to the CPM muscle progenitors as branchiomeric muscle progenitors. What about extraocular muscles, are there markers for those in the data?

We have now added references to the branchiomeric muscles. Extraocular muscles are not analyzed in the manuscript. *Pitx2*, involved in extraocular muscle development is however expressed in myoblast clusters at day 11.

Line 69: include Pax7 somewhere, since its used later to mark somitic muscle precursors. Pax7 has been included.

Line 84: *Ciona* should be italicized.

This has been corrected.

Line 95: Cardiopharyngeal doesn't need to have “C” in a capital letter. It's in the middle of the section title.

This has been corrected.

Point-by-point response to the reviewers' comments

REVIEWERS' COMMENTS

Reviewer #1 (Remarks to the Author):

The manuscript by Argiro and colleagues has significantly improved following the recent revisions, with enhanced quality and depth of data. The claims are now supported by clearer experimental evidence and quantifications. Most revision points have been addressed, making the manuscript in my opinion suitable for acceptance. However, there is one concern that I still find in the current version of the manuscript, and which was not fully addressed by the revision.

The authors assert an optimization of the protocol (lines 121-126) compared to previous publications, yet no significant new steps are specified. This could mislead readers, so I recommend revising this sentence to accurately reflect the changes.

In the revised manuscript, we have revised the sentence according to the reviewer's comment: "To generate gastruloids that form CPM and characterize their differentiation to cardiac and skeletal myogenic fates, we **extended** a previous protocol described by Rossi et al. 35 to culture gastruloids for an extended time, until day 11 (Fig. 1a) (see Methods)."

Additionally, while gastruloids survive longer, crucial gastruloid features seem to diminish over time, as evidenced by the loss of posterior tissues and a reduction in the elongation index (though not specifically measured). It appears the authors have extended the growth timeline, sometimes at the expense of axial elongation, favoring anteriorization. This allows for the capture of later-stage differentiation events. I thus suggest that the optimization claim is removed, and emphasis is placed on the need for a longer timeline for differentiation, even if it compromises gastruloid morphology.

In the revised manuscript, we have removed the optimization claim. We have added one sentence that show that gastruloid morphology is affected: "This suggests that as gastruloids develop more cardiomyocytes, they lose their posterior side, with highly beating gastruloids being predominantly anterior. **With the extension of the protocol, gastruloid morphology is affected from day 7.**"

As a minor note, for easier comparison, it would be beneficial if the scale of the scRNAseq time course data plots were consistent across time points (Fig. 3, Supplementary Fig. 5).

We have corrected Fig3 and Supplementary Fig5. Now data plots show consistent scale across time for easier comparison.

Supplementary Figure 5. Violin plots of key marker genes over time.

Violin Plots showing expression of *Mesp1*, *Tcf21*, *Isl1*, *Ebf3*, *Myl7* and *Myf5* at day 4 (a), day 5 (b), day 6 (c) and day 11 (d).

Overall, this paper will be a valuable resource for future studies, and I look forward to future developments in lineage tracing and further differentiation, as mentioned in the authors' response.

Reviewer #2 (Remarks to the Author):

In their revised manuscripts, Argiro and colleagues provide a substantial amount of new and relevant data on their already excellent original study of co-developing cardiac and skeletal myocytes in a stem cell-derived gastruloid model. The new information provided includes new data analysis, new quantifications of different cell types in the gastruloids, and new marker analysis. These data collectively provide many helpful clarifications and resulted in an improved overall manuscript. The authors have also added additional details on some of the methodology, and they have carefully revised the writing in all sections according with the new data. Lastly, the authors have expanded their discussion to include information that will be helpful to readers with and without previous expertise on this model system. All of my comments were addressed as listed specifically below.

1. Reproducibility of cell type generation and their frequencies in mouse gastruloids:

-The revised manuscript now includes a number of quantifications of cardiac and skeletal cells in different organoids from multiple differentiations. Some of the quantifications remain at a rather high level (counting beating organoids), which is a good start and can be expanded to quantification at the cellular level (percentage of each cell type per organoid and differentiation) in future studies.

-The MyoG antibody stain for skeletal muscle cells, which was added as an entirely new assay, nicely supports the HCR analysis of the original submission. In addition to strengthening the study, this will collectively provide a comprehensive set of tools for adopters of this organoid model moving forward.

-The authors added a helpful and informative paragraph with their thoughts on potential strategies to further develop the new protocol.

2. Generation of cardiomyocyte subpopulations:

The revised manuscript contains a large amount of new phenotyping data for several cardiomyocyte subtypes, via new HCR analysis. These data suggest that indeed, there seem to be multiple cardiomyocyte sub-types present in the organoids, which closely resemble their in vivo counterparts in the mid-gestation embryo heart. The analysis of left-right identity is particularly interesting, and the lineage-tracing tools incorporated in the organoid model are innovative and state-of-the art to look at these questions.

3. Mechanism of generating 2 skeletal myocyte populations in gastruloids:

The newly performed cell cycle analysis has revealed interesting information on the seemingly unexpected separation of the two skeletal myocyte populations, indeed suggesting that cell cycle status is distinct in each. I find this interesting as it suggests that at this stage of the organoid protocol, organoids may contain different developmental stages of the skeletal myocyte lineage, and having these data available to query this further will might reveal mechanisms involved in skeletal myocyte differentiation.

Minor comments: All addressed.

Reviewer #3 (Remarks to the Author):

This is a beautiful paper that is significantly improved since the original version. In this paper they show that gastruloids that are cultured to day 11, can have both skeletal muscle and cardiac cell types, which are the 2 different trajectories of the cardiopharyngeal mesoderm. They have greatly improved the manuscript by including quantification, adding expression data from embryos, to compare to gastruloids and better explains the strengths and limitations of analyzing gene expression in gastruloids versus single cell RNA-seq.

Minor comments:

1) The 22q11.2DS model mentioned on line 42 is a *Tbx1* mutant, not a deletion mouse model, in that reference.

We apologize for the confusion. We have corrected the sentence to make it clear that we are referring to a study on *Tbx1* mutant embryos: "For example, a recent report showed improper cardiopharyngeal mesoderm progenitor development in *Tbx1* conditional mutant embryos, a widely used 22q11.2 deletion syndrome (DS) mouse model, where head muscle and heart morphogenesis were impaired (Nomaru et al. Nature Com 2021)."

2) *Tbx1* is a key gene in the CPM, and they show it is nicely expressed in Figure 2n, despite very little expression in the single cell data. They might want to indicate the percent cells expressing *Tbx1* by qRT-PCR or expression analysis.

We understand the point raised by the reviewer. It is difficult to extract a percentage of cells expressing *Tbx1* from bulk qRT-PCR but have provided a new Table (Table1) with the percentage of positive *Tbx1* cells in scRNAseq data.

Table 1: Percentage of positive cells for specific genes in scRNAseq.

	Tbx1	Mesp1	Tnnt2	Myf5
day 4	1.43%	66.08%	2.03%	0.03%
day 5	2.55%	17.77%	9.02%	0.1%
day 6	1.97%	1.06%	37.07%	0.61%
day 11	1.52%	0.17%	35.22%	1.41%

3) In Figure 5n, the + sign is hard to see for *Myf5* (f) and *Myod* (g) expression. Maybe the images can be enlarged.

We have modified the picture and proposed an enlarged and slightly modified figure that should be easier to read.